



# Improved consistency in solar-induced fluorescence retrievals from GOME-2A with the SIFTER v3 algorithm

Juliëtte C.S. Anema[1,2], K. Folkert Boersma[1,2], Lieuwe G. Tilstra[1], Olaf N.E. Tuinder[1], and Willem W. Verstraeten[3]

[1]Satellite Observations Department, Royal Netherlands Meteorological Institute, De Bilt, 3730 AE, the Netherlands
[2]Meteorology and Air Quality group, Wageningen University, Wageningen, 6700 AA, the Netherlands
[3]Royal Meteorological Institute (KMI), Ringlaan 3, Ukkel B-1180, Belgium

**Correspondence:** Juliëtte Anema (juliette.anema@knmi.nl)

**Abstract.** Space-based observations of solar-induced fluorescence (SIF) provide valuable insights into vegetation activity over time. The GOME-2A instrument, in particular, facilitates SIF retrievals with extensive global coverage and a record extending over 10 years. SIF retrievals, however, are sensitive to calibration issues, and instrument degradation complicates the construction of temporally consistent SIF records. This study introduces the improved Sun-Induced Fluorescence of Terrestrial Ecosys-

tems Retrieval (SIFTER) v3 algorithm, designed to obtain a more accurate and reliable long-term SIF record from GOME-2A for the 2007–2017 period, building upon the previous SIFTER v2. The SIFTER v3 algorithm uses newly reprocessed level-1b Release 3 (R3) data, which provides a more homogenous record of the reflectances by eliminating spurious trends from changes in level 0 to level 1 processing. This improved consistency supports detailed analysis and correction of the reflectance degradation across the SIF retrieval window (734–758 nm). To address the reflectance degradation accurately, SIFTER v3

incorporates an advanced in-flight degradation correction that accounts for time, wavelength, and scan-angle dependencies throughout the entire record. Additionally, algorithm revisions have consistently reduced the retrieval residuals by around 10 % and reduced sensitivity to water vapor absorption by better capturing the atmospheric and instrumental effects. A revised latitude bias adjustment resolves unrealistic values of GOME-2A SIF over desert areas. The SIFTER v3 dataset demonstrates improved robustness and consistency, both spatially and temporally, throughout the 2007–2017 record, and aligns closely with

independent GPP measurements from the global FluxSat and FLUXCOM-X products.

## 1 Introduction

During the process of photosynthesis, vegetation emits part of the absorbed light as fluorescence. The integrated signal of this phenomenon across the canopy is known as solar-induced fluorescence (SIF). Observations of SIF provide direct measurements of photosynthetic activity and offer critical insights into terrestrial vegetation dynamics. Recent advancements in SIF retrieval

from satellite measurements have facilitated continuous global monitoring of photosynthetic activity. This includes the retrieval from spectrometer instruments, such as SCIAMACHY (Khosravi et al., 2015; Köhler et al., 2015), GOSAT (Frankenberg et al., 2011b; Joiner et al., 2011; Guanter et al., 2012), OCO-2 (Sun et al., 2018), GOME-2 (Joiner et al., 2013; van Schaik et al., 2020), TROPOMI (Köhler et al., 2018; Guanter et al., 2021), and soon, the FLEX mission (Vicent et al., 2016). With





up to nearly daily global coverage, satellite-based SIF has emerged as a powerful tool for capturing vegetation dynamics
across various biomes (e.g. Smith et al., 2018; Gerlein-Safdi et al., 2020; Mengistu et al., 2021). Furthermore, SIF has been
shown to outperform traditional reflectance-based vegetation indices in consistently tracking the impact of disturbances on
photosynthetic carbon uptake, or gross primary productivity (GPP) (e.g. Magney et al., 2019; Wang et al., 2019; Turner et al.,
2021; Qiu et al., 2022).

SIF observations from the long-running GOME-2 instruments are particularly important for their potential to eventually
construct a climate record spanning up to over 20 years. The first of the three instruments, GOME-2A, was launched onboard
Metop-A in late 2006. SIF retrieval from GOME-2 is serendipitous and stems from the spectral overlap of the instrument with
the fluorescence range, enabling retrieval of SIF from the red (∼690 nm) and far-red (∼740 nm) fluorescence peaks (Köhler
et al., 2015; Joiner et al., 2016; van Schaik et al., 2020). The additive signal emitted as SIF leads to enhanced total upwelling,
which can be detected as an infilling of Fraunhofer absorption lines present in the incoming sunlight. The spectral resolution of
the GOME-2 instrument, approximately 0.5 nm, allows for a technique that matches modelled and observed reflectance across
a spectrum covering multiple Fraunhofer lines (Joiner et al., 2013; Köhler et al., 2015; van Schaik et al., 2020). SIF retrieved
from GOME-2 is widely used to analyse interannual variations in vegetation growth (e.g. Koren et al., 2018; Gerlein-Safdi
et al., 2020; Chen et al., 2021; Fancourt et al., 2022; Qiu et al., 2022; Anema et al., 2024). However, care should be taken
to mitigate for instrumental features that cause false spatial and temporal trends, impacting the consistency of the product
(Parazoo et al., 2019).

Reflectances from GOME-2 instruments are subject to significant instrument degradation. The degradation trends exhibit
strong wavelength and even scan-angle dependencies over time (Tilstra et al., 2012b; EUMETSAT, 2022), similar to those
found in predecessor GOME (Coldewey-Egbers et al., 2008) and SCIAMACHY (Tilstra et al., 2012a). SIF is sensitive to these
effects of instrument degradation, which cause false temporal trends (Gerlein-Safdi et al., 2020; van Schaik et al., 2020; Wang
et al., 2022; Zou et al., 2024). To address these issues, van Schaik et al. (2020) implemented a seasonal wavelength-dependent
degradation correction to the level-1 reflectance data prior to the GOME-2A SIF retrieval. Although this approach enhanced
the consistency of the Sun-Induced Fluorescence of Terrestrial Ecosystems Retrieval (SIFTER) v2 product (van Schaik et al.,
2020), persistent unrealistic trends in SIF, indicating degradation, remain (Anema et al., 2024). Moreover, globally averaged
SIF values from the SIFTER v2 product show large interannual fluctuations that do not align with other SIF or vegetation
products (Wang et al., 2022; Zou et al., 2024), further suggesting ongoing inconsistency in the previous product, especially in
the later years of the GOME-2A data record.

This study aims to improve the SIF retrieval algorithm and enhance the consistency of the GOME-2A SIF long-running
record, spanning from 2007 to 2017. To achieve this, we use the new reprocessed GOME-2 level-1b dataset, Release 3 (R3),
which ensures consistent processing and auxiliary-data throughout the record. R3's homogenous nature supports the devel-
opment of degradation coefficients that accurately reflect the characteristics of degradation trends in GOME-2A reflectances.
We implement an advanced daily in-flight degradation correction that corrects for significant time, wavelength and scan-angle
dependencies identified in the level-1b reflectance data, following the methodology of Tilstra et al. (2012a, b). This correction
is applied to the level-1b R3 reflectances across 2007–2017, which then serve as input for the SIF retrieval. Additionally, we



introduce improvements in the retrieval algorithm that address atmospheric effects and incorporate enhanced understanding of

the instrument's behavior. Finally, we evaluate the consistency over time of our new SIFTER v3 dataset with SIFTER v2, the previous algorithm version, alongside independent GPP data from the FluxSat and FLUXCOM products.

## 2   GOME-2 instrument

GOME-2 is a spectrometer that measures the earthshine radiance and solar irradiance over the wavelength range 240–790 nm in four main spectral channels. Operating with a whiskbroom scanning scheme, it uses a scan mirror to perform measurements

in a nadir orbit swath with a default width of 1920 km, which enables global coverage in 1.5 days (Munro et al., 2016). For most of its orbit, the instrument scans from east to west and measures 24 forward pixels with a resolution of $80 \times 40$ km$^2$ (across $\times$ along track), followed by 8 larger back-scan pixels with a resolution of $240 \times 40$ km$^2$ for the main spectral channel data. At least once per day, GOME-2 switches to Sun mode for calibration purposes and measures the solar spectrum.

The instrument is part of the payload of the Metop satellite series (Metop-A, Metop-B, and Metop-C) that fly in a sun-

synchronous orbit with equatorial crossing at 09:30 local solar time in the descending node. GOME-2A was the first instrument to be launched, onboard the Metop-A satellite, on 6 November 2006, followed by GOME-2B and GOME-2C on 17 September 2012 and 7 November 2018, respectively. The main science objective of GOME-2 is to continuously monitor ozone column densities and other atmospheric trace gases like $NO_2$, BrO, OCIO, HCHO, $SO_2$, and $H_2O$, over a long period of time (Munro et al., 2006). Additionally, the instrument specifics enable retrieval of SIF from the near-infrared channel, which spans from

593 to 791 nm, overlapping with the far-red fluorescence peak at 740 nm (Joiner et al., 2016; Sanders et al., 2016; van Schaik et al., 2020). This channel, band 4, has a spectral resolution of $\sim$0.5 nm (full width at half maximum) and a spectral sampling interval of $\sim$0.2 nm. In this work, we focus on the SIF retrieval from the GOME-2A instrument.

Since July 15, 2013, GOME-2A operates with a reduced swath width of 960 km, resulting in ground pixels of radiance and irradiance observations measuring $40 \times 40$ km$^2$. This reduction changed the instrument's viewing geometry, from 0–53.8°

(before) to 0–33.6° (after), requiring GOME-2A data collected before and after July 15, 2013, to be treated as if recorded by two different sensors. From its early life, GOME-2A has experienced transmission loss or radiometric degradation that shows scan-angle and wavelength dependencies (EUMETSAT, 2009; Dikty and Richter, 2011; Tilstra et al., 2012b). Likely sources of the degradation have been identified as contamination on the scan mirror, along with other factors such as thin layer deposits on the detectors and degradation of the diffuser plate (Munro et al., 2016; EUMETSAT, 2022). The build-up contamination

on the exposed scan mirror was expected based on the experience with predecessor GOME (Coldewey-Egbers et al., 2008). Although the source of the contaminant is unidentified, outgassing of the satellite itself is a suspected cause (Krijger et al., 2014; Hassinen et al., 2016).

To understand and act on the noted throughput loss, EUMETSAT proceeded two throughput tests in January and September 2009. These tests, particularly the second one, altered the scan-angle related differences in the degradation rate and resulted in a

drop of throughput and a slowdown of the degradation (EUMETSAT, 2022). The different degradation trends of earth radiance and solar irradiance paths affect GOME-2A's reflectance, which represents the ratio between the two, with observed scan-angle



dependencies across UV and NIR ranges (Tilstra et al., 2012b). We will further discuss the characteristics of the reflectance degradation in the NIR range in Sect. 3. The reflectance degradation impacts the retrieval of various level 2 products, including the absorbing aerosol index (Tilstra et al., 2012b), ozone profiles (Cai et al., 2012) and SIF (van Schaik et al., 2020).

The long-term consistency of the reflectance measurements is affected not only by changes at instrument level, such as degradation, but also by variations in raw data processing to level-1b data (EUMETSAT, 2022). The level-1b data contains calibrated earth radiance and solar irradiance measurements, as well as auxiliary information such as geolocation and cloud parameters. Processor-related variation in the level-1b data can introduce spurious trends and complicate data coherence over time, which is crucial for climate monitoring studies. Inconsistent processing over time raised concerns in previous SIF re-

trievals from GOME-2A, given the retrieval's sensitivity to small changes in level-1b data (van Schaik et al., 2020). To address this issue, EUMETSAT has recently reprocessed the GOME-2A level-1b dataset using the level 0-to-1b operational processor version 6.3.3 across the entire 2007–2018 period. By eliminating processor-version related changes in the dataset, a more homogenous dataset is achieved, facilitating the evaluation and construction of soft calibration corrections to mitigate long-term degradation of the instrument (EUMETSAT, 2022).

Figure 1 shows the time series of monthly averaged reflectance (at 740 nm) over the Sahara between 2007 and 2017. Although GOME-2A was in operation till 2021, we omit data beyond January 2018 due to the loss of orbit control for Metop-A, resulting in a satellite drift and constrained solar visibility of GOME-2A. The reflectance data are presented for two data sets: (i) the reprocessed level-1b data using processor version 6.3.3, and (ii) the previous level-1b data used in SIFTER v2, which encompasses three processor version (5.3, 6.0, and 6.1) over time. In this work we will use the reprocessed R3 level-1b

data to retrieve SIF.

The difference in reflectance magnitude between both datasets is small and diminishes progressively with newer processor versions in the previous level-1b dataset, as seen from Fig. 1. Thus, the differences between successive processor versions become smaller, indicating that the data processed with v5.3 differs more from R3 data than data processed with v6.1. Overall, this suggests improved temporal consistency in reflectance due to the reprocessing. Furthermore, both datasets show similar

tendencies with noted degrading trends, particularly dominant in the later period after 2014, and a strong jump after the sensor switch change in July 2013. This jump is caused by the change in viewing zenith angles after the swath reduction.





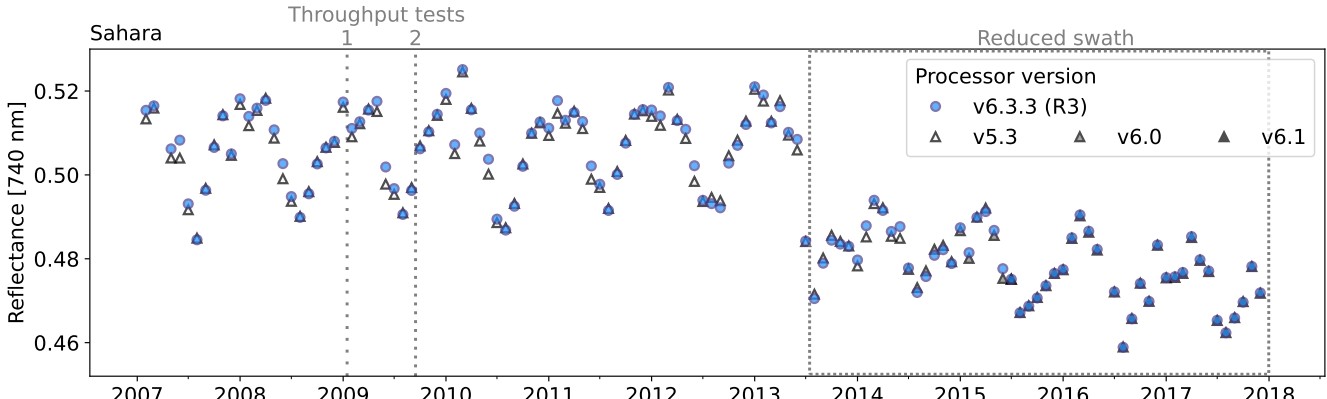

**Figure 1.** Monthly averaged reflectance measured by GOME-2A at 740 nm over the Sahara region (16–30° N, 8° W–29° E) as a function of time. Reflectances ($R(\lambda)$) are obtained by: $(\pi I(\lambda))/(I_0(\lambda)\cos(\theta_0))$, where $I(\lambda)$ is the radiance, $I_0(\lambda)$ the solar irradiance and $\theta_0$ the solar zenith angle. Reflectances obtained from the re-processed R3 level-1b data have processor version 6.3.3 and are shown in blue bullets. Reflectances of the previous level-1b data set consists of different processor versions (v5.3, v6.0, and v6.1) and are shown in triangles with different hues of grey. The reflectances from both level-1b datasets were co-sampled, and both data had to meet the criteria of cloud fraction <0.4. Data of days that are listed as outliers, including measurements in narrow swath, are excluded from both datasets. Events that impact the reflectance are indicated by grey dashed lines.

## 3 In-flight degradation correction

In this section, we describe a method to analyze and address the scan-angle and wavelength dependency of degradation trends observed in the near-infrared GOME-2A reflectances. Adapted from the method used to correct for reflectance degradation in SCIAMACHY and GOME-2A for absorbing aerosol index (AAI) retrieval (Tilstra et al., 2012a, b), our in-flight degradation correction method examines daily global mean reflectance over time ($t$, per day), scan-angle position ($s$) and wavelength ($\lambda$).

This more advanced approach improves upon the degradation correction used in SIFTER v2 (van Schaik et al., 2020), which relied on a wavelength-dependent seasonal factor with the 2007–2012 reflectance average as baseline. Furthermore, the correction was only applied to reflectances from June 2014 onward (see Table 2). In contrast, our method applies a continuous daily correction factor across the entire 2007–2017 record that addresses the early degradation patterns and the varying trends of degradation over time (e.g. before and after the throughput tests). Moreover, we include a scan-angle dependency into the degradation correction. Findings from Tilstra et al. (2012b) and EUMETSAT (2022), which identified scan-angle dependencies in GOME-2 reflectance – particularly in shorter wavelengths and in eastward-looking directions – prompted the inclusion.

We analyzed the global mean reflectances at each measured wavelength between 712 and 785 nm (in total 356 detector pixels), covering the SIF retrieval window (734–758 nm) and the spectral bands used to calculate the scene albedo (van Schaik et al., 2020). Valid reflectances of GOME-2A ($R_{\lambda,s}$) are collected between 60°S and 60°N, with solar zenith angles smaller than 85°, and averaged daily. Observations containing sun glint or cloudy conditions are not filtered out, but days corresponding



to narrow swath and nadir static measurements are excluded from the analysis. For GOME-2A, the scan position parameter $s$ runs from 1 (eastward) to 24 (westward), considering only forward pixels. The daily global reflectance at 747.1 nm for the

most eastward ($s$=1), center ($s$=12), and westward ($s$=24) pixels from 2007 to 2012 is shown in Fig. 2.

We modeled the variation in daily global mean reflectance ($R^*_{\lambda,s}$) using a Fourier series ($F^q_{\lambda,s}$), representing the seasonal variation, multiplied by a polynomial term ($P^p_{\lambda,s}$) to account for instrument degradation over time:

$$R^*_{\lambda,s} = P^p_{\lambda,s}[1 + F^q_{\lambda,s}] \qquad (1)$$

Here, $p$ denotes the degree of polynomial $P_{\lambda,s}$, and $q$ represents the order of the finite Fourier series $F_{\lambda,s}$. The implicit

assumption is that the global averaged reflectance shows no long-term trends. Therefore, the observed trends captured by the polynomial are attributed to instrumental effects.

We analyzed $R^*_{\lambda,s}$ separately for the period before and after the sensor switch on 15 July 2013. Equation (1) is fitted to reflectances collected between (i) January 2007 and December 2012, and (ii) between January 2007 and December 2017. In the latter fit, reflectances between 2007 and July 15, 2013, were first interpolated to match the scanning angles covered by the

reduced swath. Each fit covers complete years to prevent biases from seasonal variation. Here, we set $p$=2 for the 2007–2012 fit, and $p$=3 for the 2007–2017 fit. These higher order polynomial degrees offer the flexibility to capture the varying behavior of the instrument degradation over time (discussed in Sect. 2). In both fits, $q$=6 was used for the order of the Fourier series. Table 1 summarizes the characteristics of the 2007–2012 and 2007-2017 fits.

The resulting 2007–2012 fits of $R^*_{\lambda,s}$ for the most eastern ($s$=1), center ($s$=12) and western ($s$=24) scan-indices are shown as

solid black lines in Fig. 2. These fits accurately follow the temporal pattern of the global reflectance, with correlations of 0.99, 0.99, and 0.98, respectively, for the scan-indices 1, 12, and 24. The dashed lines represent the fitted polynomial component $P^p_{\lambda,s}$, illustrating the degradation effect over time. A gradually increasing trend is noted in the 747.1 nm reflectance, particularly at the most eastward position ($s$=1). This increase in reflectance results from the stronger decrease in measured solar irradiance (the *denominator* in $R$) than in measured radiance (the *numerator* in $R$). Furthermore, a slight reduction in increase is noted

around 2010, possibly the result of the second throughput test in September 2009 (Fig. 1) (Munro et al., 2016; EUMETSAT, 2022).




**Table 1.** Summary of the characteristics for the fitted global mean reflectance over 2007–2012 and 2007–2017. Collected reflectances have latitudes between 60°S and 60°N and solar zenith angles < 85°. The scanning angles reflect the instrumental mirror angle, with ranges around -45.5–45.4° for the 1920-km swath, and around -28.3–28.3° for the reduced 960-km swath (post-15 July 2013). Fitting parameters $p$ an $q$ represent the polynomial degree of $P_{\lambda,s}$ and the order of the finite Fourier series $F_{\lambda,s}$ in Eq. (1). The derived correction factors are applied to reflectances at 712–785 nm and scan-index 1–24 to the specified application periods.

| Fit period | Scanning angles [°] | Fit parameters | | Application period |
|---|---|---|---|---|
| | | $p$ | $q$ | |
| 4 Jan. 2007 – 31 Dec. 2012 | -45.4 – 45.4 | 2 | 6 | 4 Jan. 2007 – 15 July 2013 |
| 4 Jan. 2007 – 31 Dec. 2017 | -28.3 – 28. 3 | 3 | 6 | 16 July 2013 – 31 Dec. 2017 |

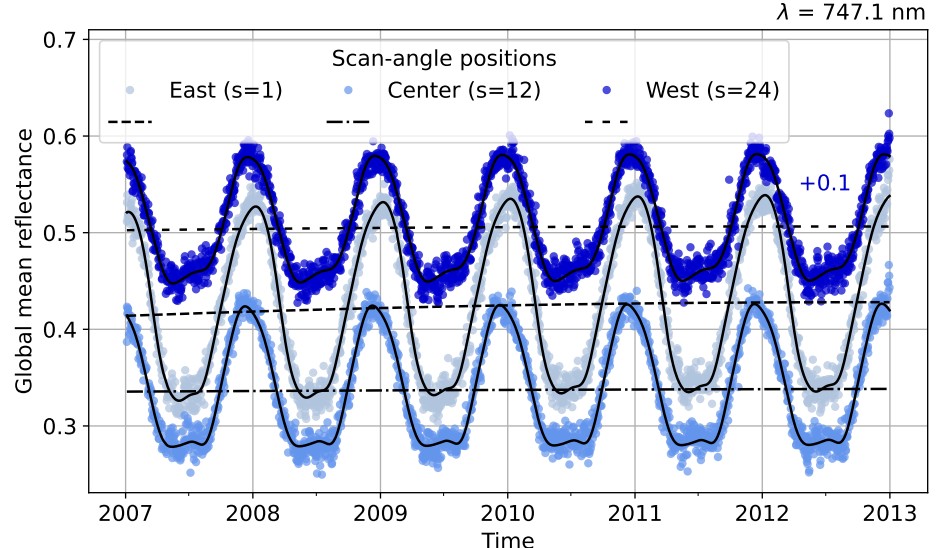

**Figure 2.** Global mean daily reflectance at 747.1 nm from GOME-2A over time, between 2007 and 2012, for the most eastward ($s$=0), center ($s$=12), and most westward ($s$=24) scan-angle positions. Differences in magnitude and amplitude of the plotted reflectance time series are caused by the angular dependences of cloud and surface reflection. To separate the time series graphically, an offset of 0.1 was added only to the reflectances of the most westward scan-angle position ($s$=24) (denoted in dark blue). The solid black lines show the fitted daily global reflectance $R^*_{\lambda,s}$ and the dashed black lines illustrate the effect of instrument degradation, as described in the main text.

Finally, the correction for instrument degradation is obtained by scaling the polynomial value at each day to its value at the reference date ($t_0$):

$$c_{\lambda,s}(t) = P^p_{\lambda,s}(t_0)/P^p_{\lambda,s}(t) \tag{2}$$

The reference date for both fits, 2007–2012 and 2007–2017, is January 5$^{\text{th}}$, 2007. We use the polynomial from the 2007–2012 fit to derive the correction factors for January 2007 to mid July 2013, and that of the 2007–2017 fit for mid July 2013 to



December 2017 (Table 1). We multiply these wavelength and scan-angle dependent corrections by the level-1b reflectances to correct for the degradation trends.

Figure 3 shows the averaged correction factor at 747.1 nm, broken down per year and scanner angle, for the observations (a) before and (b) after the sensor-switch. In both periods, the variation in correction factors across scan-angle positions is of the same order as the variation over time, reaching up to 5 %. This indicates the significance of a scan-angle dependent degradation correction. From 2007 to 2014 the annual averaged correction factors at 747.1 nm are below one, meaning the reflectances were higher than the baseline in January 2007. From 2015, the correction factor for most scan indices exceeds one and then gradually increases further to correct for the decreasing reflectances over time.

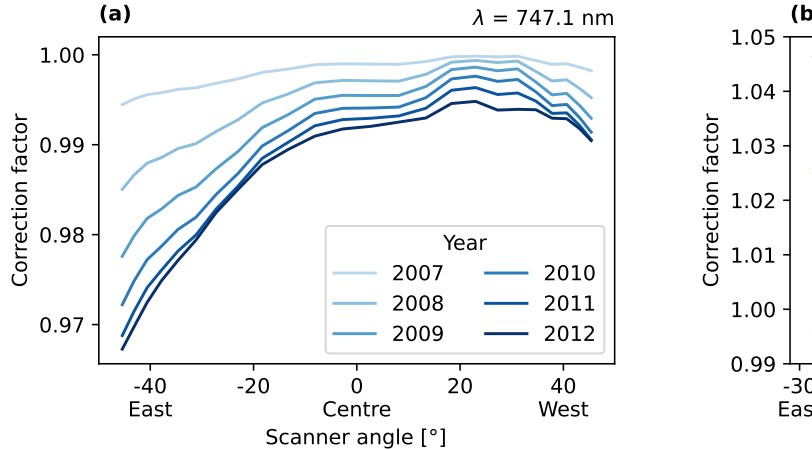
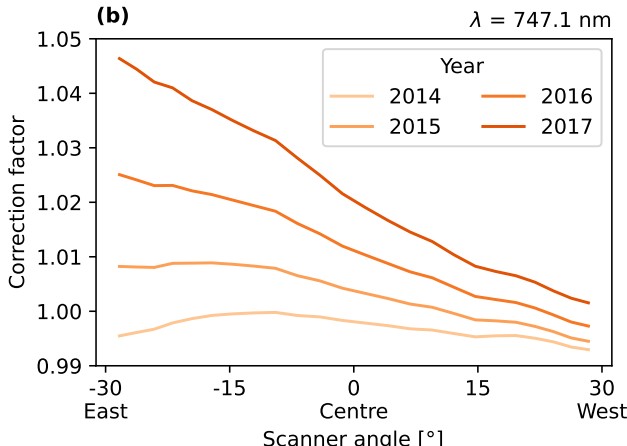

**Figure 3.** Reflectance degradation correction factors at wavelength 747.1 nm, averaged annually across the 24 scan-angle positions. Panels (a) and (b) show the obtained corrections over the periods before and after the sensor-switch at the 15[th] of July 2013, respectively. Only full-year averages are shown, so 2013 is omitted due to the mid-year sensor switch.

The gradual shifts in the correction factors over time are also visible in Fig. 4, which displays the factors over time and wavelength across the retrieval window range (734–758 nm) at scanning angle -28.3° ($s$=1 for post-sensor switch period). The correction factors reflect the gradual increase of the reflectance from 2007, followed by a tipping point around 2011–2012 for most wavelengths, ultimately falling below the baseline of January 2007. This transition from lowering ($c$<1) to increasing corrections ($c$>1) occurs around 2014–2015, with this shift occurring earlier at shorter wavelengths.

Our analysis shows strong variations in reflectance over time, with increases around 2007–2012 followed by decreases from around 2014. These variations arise from the changing rates of throughput loss in both the Earth and solar paths, where the relative rates of throughput loss in these paths determine the observed fluctuations in reflectance. Overall, the impact of instrument degradation on the measured GOME-2 reflectance in the NIR show a parabolic pattern with time, a stronger effect in the eastward scan positions, and a decrease with increasing wavelengths.





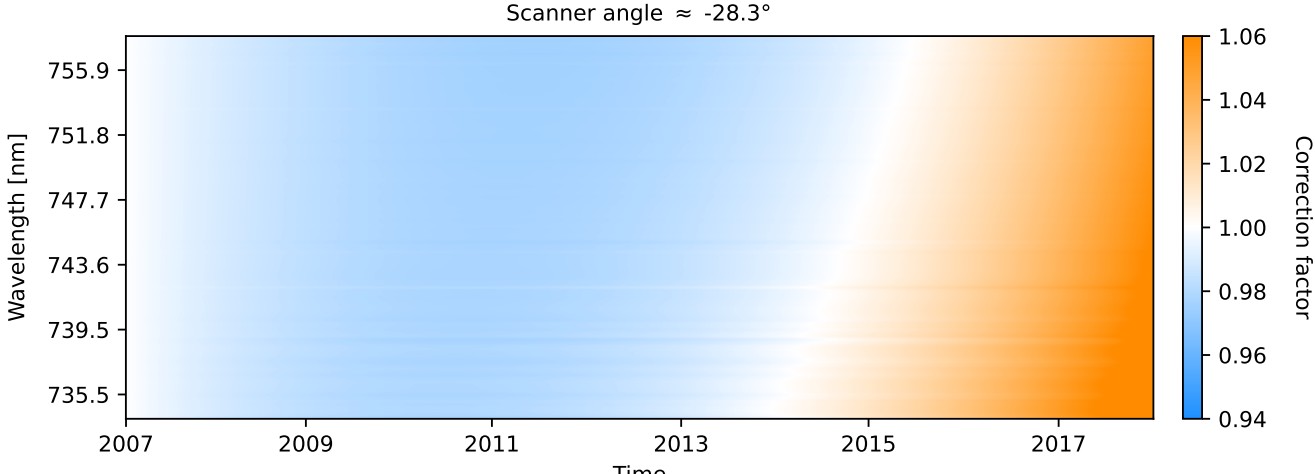

**Figure 4.** Temporal evolution of the degradation correction factors for each wavelength within the SIF retrieval window (734–758 nm, n=118) at scanner angle -28.3°, representing the easternmost scan position, $s$=1, in the reduced-swath mode (similar to $s$=7 in the 1920-km swath). The shown correction factors are derived from the 2007–2017 reflectance fits.

## 4  SIFTER v3 retrieval algorithm

### 4.1  SIFTER retrieval algorithm

The SIFTER retrieval algorithm relies upon the relative filling-in of solar Fraunhofer absorption lines by the far-red SIF (peak at ∼740 nm) – an approach pioneered by Joiner et al. (2011), Frankenberg et al. (2011b), and Guanter et al. (2012) for SIF retrieval from space. To detect the in-filling of the narrow Fraunhofer lines from GOME-2A, with a spectral resolution of 0.5 nm, a spectral window covering multiple deep Fraunhofer lines is needed (Parazoo et al., 2019). Since version 2, the SIFTER algorithm uses a relatively narrow retrieval window of 734–758 nm to limit complications by absorption signatures from water vapour and the O2-A band (van Schaik et al., 2020). The key of the retrieval is to separate the contributions of surface-emitted SIF from those of direct reflection to the overall reflectance. This is done by matching a modelled spectrum to the measured reflectance. Using a Lambertian model, the reflectance ($R$) can be written as:

$$R(\lambda, \mu, \mu_0) \approx a_s(\lambda) T^{\downarrow}(\lambda, \mu_0) T^{\uparrow}(\lambda, \mu) + \frac{\pi I_{\text{SIF}}(\lambda)}{E_0(\lambda)\mu_0} T^{\uparrow}(\lambda, \mu) \tag{3}$$

where $\mu$ and $\mu_0$ are the cosines of the viewing and solar zenith angles, respectively. The atmospheric molecular scattering in the near-infrared region is neglected (e.g. Joiner et al., 2013). In Eq. (3), $a_s$ is the surface albedo, $T^{\uparrow}$ and $T^{\downarrow}$ are the upward and downwelling atmospheric transmission factors, respectively, and $I_{\text{SIF}}$ is the SIF emission at the Earth's vegetated surface. The SIFTER algorithm uses a fourth-order polynomial to describe the spectral surface albedo $a_s$. The atmospheric transmittance $(T^{\uparrow}, T^{\downarrow})$ is described by 10 principal components (PCs) that are obtained over non-vegetative areas.



The 10 basic functions describing the atmospheric transmittance are obtained in three main steps. First, a large ensemble of reflectance spectra without cloud coverage (cloud fraction < 0.4) are collected over a non-vegetative reference area to represent the transmittance for a wide variety of conditions where $I_{\mathrm{SIF}}$ equals zero. The Sahara region (16–30 °N, 8 °W–29 °E) is used as reference area and spectra are collected over the January 2007 to December 2012 period. Secondly, the contribution of the surface albedo to the spectra is eliminated by subtracting a second-order polynomial function obtained over spectral regions with negligible atmospheric absorption over no cloud conditions (described in more detail in van Schaik et al. (2020)). Finally, using the widely used iterative principal component analysis NIPALS (Nonlinear Iterative Partial Least Squares) (Esbensen et al., 2002), the first 10 principal component spectral functions ($f_k(\lambda)$) are retrieved that explain the variance within the spectra. Note that for the retrieval of data after the sensor switch in July 2013, PC's are obtained from spectra with viewing zenith angles < 35° to match the observations in reduced swath.

Following the preparatory step of obtaining the 10 PCs, the SIFTER algorithm proceeds to fit a model to each individual reflectance spectrum by minimizing the differences between the observed and modeled reflectance spectra using a Levenberg-Marquardt least-squares regression (van Schaik et al., 2020). The modeled spectra ($R_m$), follows on Eq. (3) and is written as:

$$R_m(\lambda, \mu, \mu_0) \approx (\sum_{j=1}^{n} a_j \lambda^j) e^{-\sum_{k=1}^{m} b_k f_k(\lambda)} + \frac{\pi c I_{\mathrm{SIF}}(\lambda)}{\mu_0 \overline{E_0}} e^{-\frac{\mu^{-1}}{\mu^{-1}+\mu_0^{-1}} \sum_{k=1}^{m} b_k f_k(\lambda)} \tag{4}$$

Here the $a_j$ represent the fitting coefficients of the surface albedo, the $b_k$ are the coefficients that match the set of the 10 PCs, $f_k(\lambda)$, describing the atmospheric transmission contribution to $R$, and $c$ is the fit coefficient for $I_{\mathrm{SIF}}$ that characterizes the strength of the fluorescence signal. In total there are 16 fitting coefficients: 5 from $a_j$, 10 from $b_k$, and 1 from $c$. In Eq. (4), the solar irradiance, $\overline{E_0}(\lambda)$, is taken to be the solar irradiance reference spectrum from Chance and Kurucz (2010) convolved with the GOME-2A slit function and scaled to the Earth-Sun distance. The solar irradiance reference spectrum is used in the model to mitigate the influence of instrument degradation on the observed solar irradiance (Sanders et al., 2016).

## 4.2 Processing and improvements of SIFTER v3 retrieval

The GOME-2A SIFTER v3 product presented here is an implementation of the general approach described in Sect. 4.1 and the proposed in-flight degradation correction described in Sect. 3. For the SIF retrieval process, EUMETSAT's reprocessed level-1b R3 data was used as input. The daily degradation correction factors were applied to stabilize the R3 reflectances over the entire 2007–2017 record. These degradation corrected reflectances were then used to extract the 10 Principal Components (PCs) that describe the atmospheric transmission, and to retrieve SIF. Figure S1 shows a flowchart of the processing algorithm.





**Table 2.** Overview of the implemented changes, ordered by importance for the fit retrieval, in the SIFTER algorithm by this study: a comparison between SIFTER v3 and SIFTER v2 (van Schaik et al., 2020). The changes are placed in order of largest impact on the fit retrieval. Changes (1) and (2) are expected to significantly affect the fit quality of the retrieval, whereas alteration (3) is expected to mainly impact the consistency of SIF over time and across scan-angles.

| | Subject of change | SIFTER v2 | SIFTER v3 |
|---|---|---|---|
| 1 | Scaling method of level-1 reflectance reference spectra prior to PCA | Variance scaling | Standard deviation scaling |
| 2 | Interpolation across slit functions for convolution with reference solar spectrum | Fully-sampled pixels (n=10) | Both fully and not fully-sampled pixels (n=765) |
| 3 | Degradation correction | In-flight degradation correction | In-flight degradation correction |
| | • Time step | • Seasonally | • Daily |
| | • Scan-angle dependent | • No | • Yes |
| | • Baseline | • 2007–2012 | • January 5, 2007 |
| | • Application period | • From June 2014 | • 2007–2017 |

In SIFTER v2, the collected level-1 reflectance spectra over the reference area (Sahara) underwent mean centering and scaling by the variance across wavelengths before conducting the principal component analysis (PCA). Our study explores an

225 alternative 'autoscaling method', that combines mean centering with scaling by the standard deviation, which ensures equal importance across the variables (van den Berg et al., 2006). Figure S3 demonstrates that the use of standard deviation scaling on the mean-centered spectra results to a more distinct pattern with tighter magnitude variations across the wavelengths as compared to variance scaling. This autoscaling approach outperforms the SIFTER v2 pretreatment method, with the 10 PCs explaining 99.95 % of the spectra variance compared to 99.85 %, as shown in Fig. 5(c). More orthogonal PCs are expected to

230 improve the fit by capturing the variation across wavelengths more consistently, leading to a more accurate representation of the atmospheric transmission.





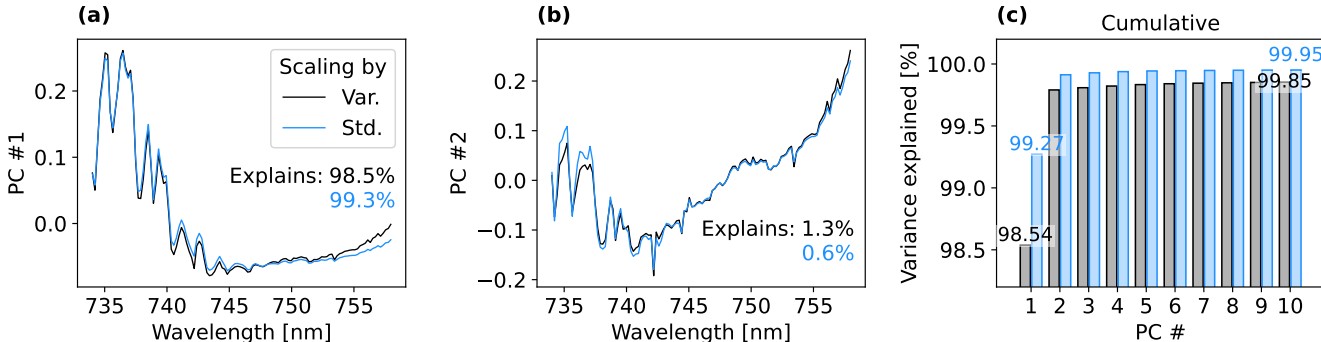

**Figure 5.** The leading principal components, (a) PC 1 and (b) PC 2, of GOME-2A reflectance spectra using scaling by the standard deviation (in black) and the variance (in blue) as pre-treatment method prior to the principal component analysis. Panel (c) shows the cumulative explained variance of PCs 1 to 10 using both pre-treatment scaling methods. The PCs are obtained from collected GOME-2 reflectance spectra over the Sahara between 2007 and 2012.

Figure 6 shows the spectral fit over a single Congo Basin pixel with PC's obtained when using (i) the variance as scaling factor as in SIFTER v2, and (ii) the standard deviation as a scaling factor. All other settings and input of the retrieval for both tests are equal and follow that of SIFTER v3. The use of the standard deviation scaling factor results in lower SIF (0.99 mW m$^{-2}$ sr$^{-1}$ nm$^{-1}$), compared to the variance scaling factor (1.07 mW$^{-2}$ sr$^{-1}$ nm$^{-1}$). Furthermore, PC$_{std}$ resulted in more homogenized and lower fit residual across the spectrum, as seen by the RMSE (Fig. 6b). Notably, PC$_{var}$ had more difficulty to explain the 734–745 nm region of the spectrum (RMSE of 0.089 %), where water absorption features are more prominent, in comparison to the PC$_{std}$ (RMSE of 0.069 %) over the same spectral region. The observation of reduced fit residuals when PC$_{std}$ are used, specifically over 734–745 nm, holds true across a larger ensemble of pixels (Congo Basin, n=644) as illustrated in Fig. S4. This confirms that the proposed autoscaling pretreatment method results in PCs that better describe the atmospheric transmission (including the water vapor absorption), resulting in better fit of the retrievals.





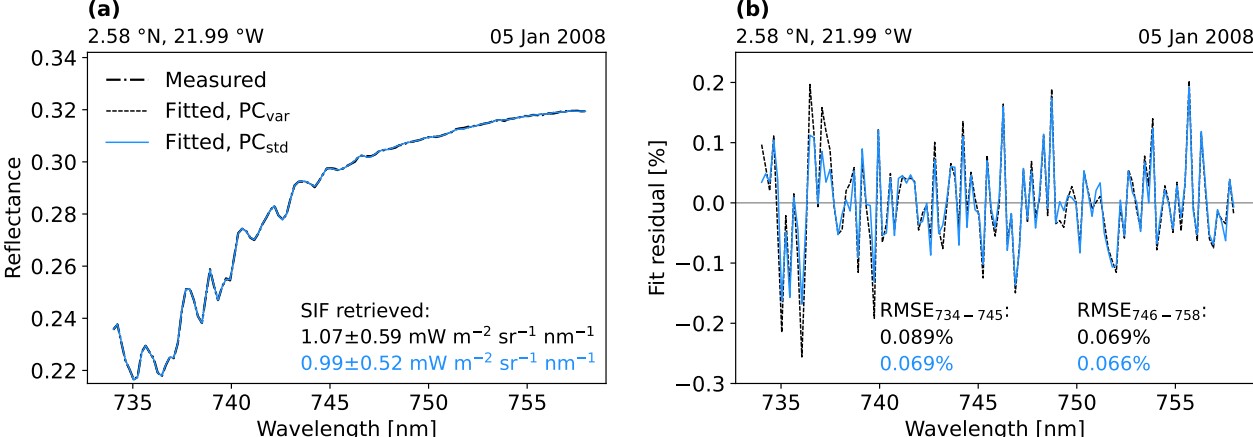

**Figure 6.** SIF retrieval from GOME-2A of a single pixel over the Congo Basin (2.58 °N, 21.99 °W) on 5 January 2008 using PCs computed with standard deviation and variance as pretreatment scaling factors, respectively referred to as $PC_{std}$ and $PC_{var}$. Panel (a) shows the observed reflectance and fitted reflectance for both retrievals and (b) shows the fit residuals of the retrievals. The stated SIF values reflect the values from the retrieval step and are not corrected for the latitude bias.

We now turn to the revision of the convolution of the slit function with the solar irradiance reference spectrum. In the SIFTER algorithm, the high-resolution solar irradiance reference spectrum from Chance and Kurucz (2010) is used, convolved with the GOME-2A slit function, and scaled to the Earth-Sun distance (Sanders et al., 2016; van Schaik et al., 2020). We denote this spectrum with $\overline{E_0}$. Before this convolution can be done, we interpolate the slit functions across the wavelengths to match the high spectral resolution of the reference spectrum ($\Delta\lambda$=0.01 nm). We use slit functions in the range of ∼612–770 nm. In SIFTER v2, only the slit functions from fully-sampled detector pixels (n=10) are used for the interpolation process, encompassing two spectral points within the retrieval window (Fig. S5). In this work, we include all slit functions provided in the key-data (n=765), consisting both of fully-sampled and not-fully sampled pixels, which are interpolated by EUMETSAT as described in Siddans et al. (2012). Figures S6a–d present the slit functions and their interpolation to the 0.01 nm spectral resolution, following the methods of SIFTER v3 and SIFTER v2, respectively.

Figure 7 shows the reference solar spectrum convolved with the slit functions as done in SIFTER v2 and in SIFTER v3 (this work). The solar irradiance spectrum convolved using the expanded slit function dataset follows the fine features of the Fraunhofer absorption lines better, resulting in deeper Fraunhofer lines than the convolved spectrum using only the fully-sampled pixels.





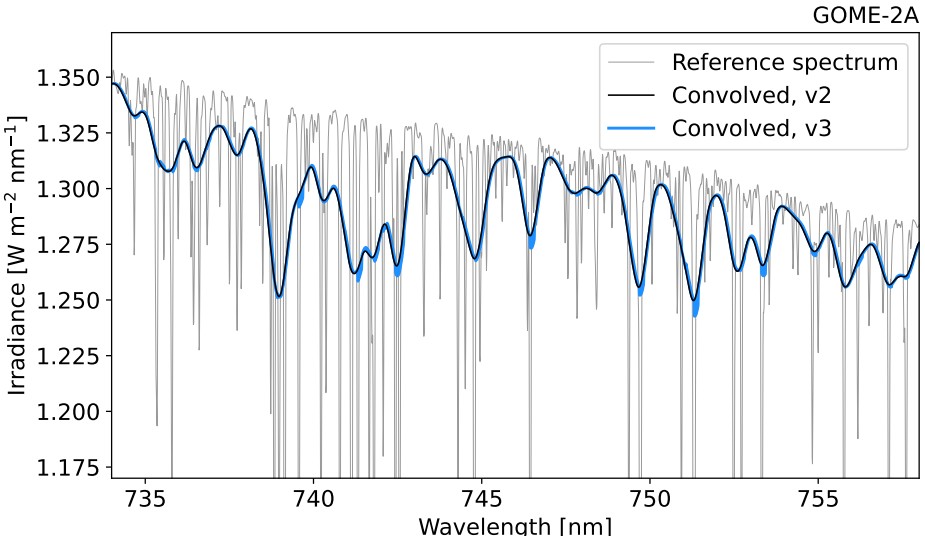

**Figure 7.** The solar irradiance reference spectrum of Chance and Kurucz (2010) convolved with the GOME-2A slit function (in grey), and the Chance and Kurucz (2010) spectrum convolved with the interpolated SIFTER v2 slit function (using fully-sampled pixels) (in black), and the interpolated SIFTER v3 slit function (using fully and not-fully sampled pixels) (in blue).

The solar irradiance reference spectrum $\overline{E_0}$ is applied at two instances in the algorithm retrieval. Firstly, the spectrum is used for high sampling interpolation of the measured solar irradiance spectrum to match the wavelength grid of the radiance measurements for the reflectance calculation. Secondly, the $\overline{E_0}$ spectrum is used in the modeled reflectance, Eq. (4), replacing the observed solar spectrum $E_0$, as discussed in Sect. 4.1. To understand the impact of the changes in the solar irradiance

reference spectrum on the SIF retrieval fit, we examine the impact on both individual usages. Figure 8 presents the fitted reflectance and the fit residual on the single Congo Basin pixel for retrievals with input reflectances that use high sampling interpolation obtained from the solar irradiance reference spectrum $\overline{E_0}$ as in (i) SIFTER v2 (HSI$_{v2}$) and in (ii) SIFTER v3 (HSI$_{v3}$). The HSI$_{v3}$ reflectances resulted in a reduction of SIF by 15 % in comparison to the HSI$_{v2}$. Additionally, employing all key-data did result in a slightly better fit as seen from the reduction in fit residuals. This showcases the strong sensitivity of

the SIF retrieval algorithm to slight changes in the reflectances.

Figure 9 shows the fit results and residuals for the use of $\overline{E_0}$ as obtained in SIFTER v2, $\overline{E_{0,v2}}$, and SIFTER v3, $\overline{E_{0,v3}}$ on the single Congo Basin pixel. The retrieved SIF slightly increased from 0.89 mW m$^{-2}$ sr$^{-1}$ nm$^{-1}$ to 0.99 mW m$^{-2}$ sr$^{-1}$ nm$^{-1}$ when $\overline{E_{0,v3}}$ is used instead of $\overline{E_{0,v2}}$ in the modeled reflectance. This increase results from the deeper Fraunhofer absorption lines in the solar irradiance reference spectrum convolved with the slit function obtained using the SIFTER v3 method (see

Fig. 7).





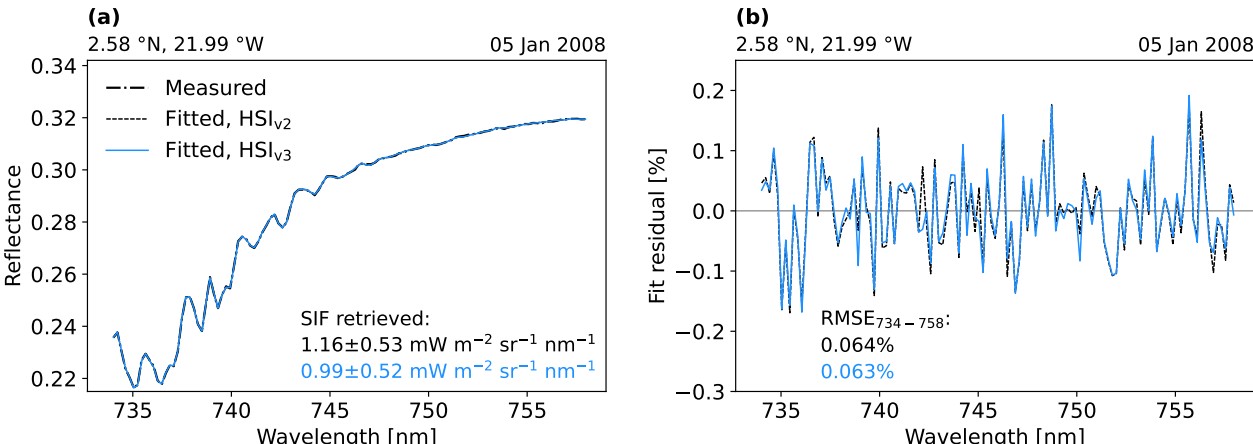

**Figure 8.** Similar to Fig. 6, the graphs show the (a) fitted reflectance and (b) fit residuals for retrievals from reflectances obtained using the SIFTER v3 and SIFTER v2 wavelength calibration, referred to as $HSI_{v3}$ and $HSI_{v2}$ respectively.

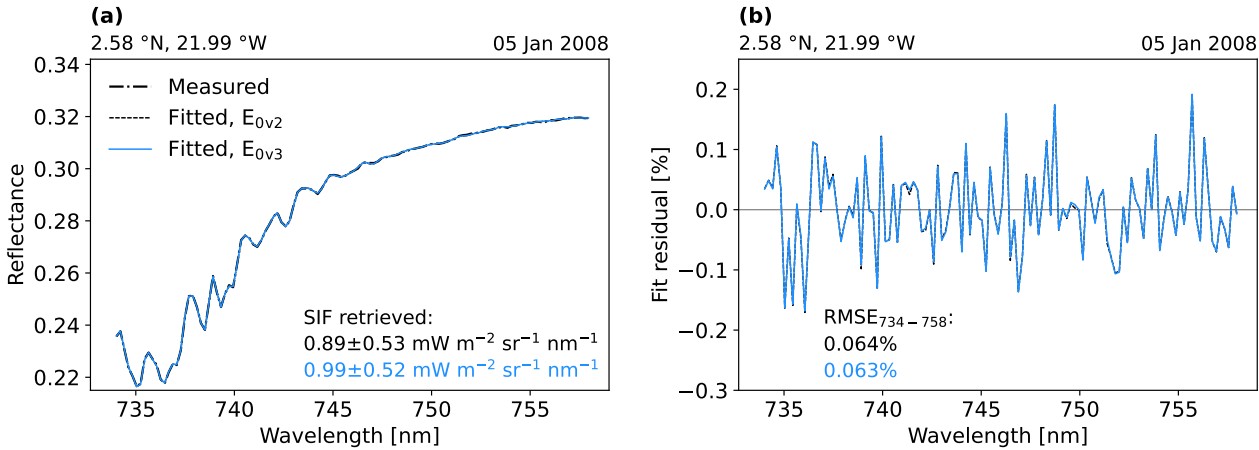

**Figure 9.** Similar to Fig. 6, the graphs show the (a) fitted reflectance and (b) fit residuals for retrievals with the SIFTER v3 and SIFTER v2 convolved reference solar spectra as the solar irradiance $E_0$, referred to as $E_{0,v3}$ and $E_{0,v2}$ respectively.

To study the effects of the algorithm revisions on the retrieval performance in tandem, and for a larger ensemble of pixels, we conducted a series of four experiments in which the revisions are progressively introduced from SIFTER v2 to SIFTER v3 methodology. The first experiment is based on the SIFTER v3 algorithm approach, but with the use of (i) the variance scaling pre-treatment method ($PC_{var}$), and the solar irradiance reference spectrum convolved with GOME-2A slit function as done in SIFTER v2 ($\overline{E_{0,v2}}$) for (ii) the high sampling interpolation ($HSI_{v2}$) and (iii) computation of the modelled reflectance. The last experiment implements all revisions proposed in this work. Table 3 shows the results of the experiments over 678 pixels on 5 January 2008, over the Congo Basin (13 °S–6 °N, 14–31 °W).



**Table 3.** Summery of experiment results on the progressive revisions from SIFTER v2 (experiment 1) to SIFTER v3 (experiment 4) settings over 678 pixels in the Congo Basin (13 °S–6 °N, 14–31 °W) on 5 January 2008. The selected pixels of the four experiments were co-sampled and had to meet the requirements of autocorrelation < 0.2 and cloud fraction < 0.4. The scan-angle dependent degradation correction as implemented in SIFTER v3 was used in all experiments.

|  | Experiment | | | SIF value $\pm$ uncertainty | RMSE fit residual [%] | | |
|---|---|---|---|---|---|---|---|
|  | PC | HSI | $\overline{E_0}$ | [mW m$^{-2}$ sr$^{-1}$ nm$^{-1}$] | <746 nm | >746 nm | 737–758 nm |
| 1 | $PC_{var}$ | $\overline{E_{0,v2}}$ | $\overline{E_{0,v2}}$ | $1.44 \pm 0.58$ | 0.075 | 0.061 | 0.068 |
| 2 | $PC_{std}$ | $\overline{E_{0,v2}}$ | $\overline{E_{0,v2}}$ | $1.20 \pm 0.57$ | 0.070 | 0.061 | 0.065 |
| 3 | $PC_{std}$ | $\overline{E_{0,v3}}$ | $\overline{E_{0,v2}}$ | $1.00 \pm 0.54$ | 0.067 | 0.059 | 0.063 |
| 4 | $PC_{std}$ | $\overline{E_{0,v3}}$ | $\overline{E_{0,v3}}$ | $1.04 \pm 0.54$ | 0.066 | 0.059 | 0.063 |

The revision of the pre-treatment scaling method before the principal component had the most substantial impact on both the magnitude of SIF and the retrieval fit. The use of the $\overline{E_{0,v3}}$ showed the largest effect in terms of altering the reflectance in
its use for the wavelength calibration of the irradiance observations to match the radiance wavelength grid. The results indicate that the discussed revisions in SIFTER v3 decreased the magnitude of SIF by 28 %, from 1.44 mW m$^{-2}$ sr$^{-1}$ nm$^{-1}$ to 1.04 mW m$^{-2}$ sr$^{-1}$ nm$^{-1}$, but substantially improved the retrieval fit quality from 0.068 % to 0.063 % RMSE fit residuals. The proposed revisions decrease the sensitivity to water vapor as indicated by the reduced RMSE fit residual over the 734–746 nm where water vapor features are present. This improvement is mainly attributed to the autoscaling method in the pre-treatment
of the principal component analysis, resulting in PCs that better capture the water vapor features.

### 4.3 Zero-level offset adjustment

Our retrieval detects the presence of chlorophyll fluorescence as changes in the relative depth of the Fraunhofer lines. However, instrumental effects and artifacts can also cause in-filling, or deepening, of the Fraunhofer lines, making it indistinguishable from true fluorescence signals (e.g. Frankenberg et al., 2011a; Joiner et al., 2012; Khosravi et al., 2015). These false SIF signals
are noted as so-called zero-level offsets over vegetation-free regions and may be of the same order of magnitude as SIF (Köhler et al., 2015). Previous studies have found a particularly strong latitude dependency in GOME-2A SIF over oceans (e.g. Köhler et al., 2015; Joiner et al., 2016; van Schaik et al., 2020). This latitudinal bias is possibly related to the varying width of the slit function, driven by the changing temperatures of the instrument in the descending orbit (Munro et al., 2016). Specifically, the widening of the slit function, due to a higher optical bench temperature, may cause ever shallower Fraunhofer lines along
the orbit, which the SIFTER algorithm would interpret as in-filling by chlorophyll fluorescence, while a narrower slit function results in deeper Fraunhofer lines, leading to retrieved negative SIF values over unvegetated regions.

Such a zero-level offset is indeed visible in the new SIFTER v3 data. The effect in SIF is visualized in Fig. 10a (blue line), showing a negative trend over the Northern Hemisphere and a positive trend over the Southern Hemisphere above the Pacific Ocean, where SIF is expected to equal zero. Here we apply a *post hoc* correction to adjust for the observed bias. The applied
adjustment was adapted from the method of SIFTER v2. In this method, van Schaik et al. (2020) obtained day and latitude-band



specific regression coefficients between collected reflectance (at 744 nm) and SIF data per 1° latitude band over the Pacific Ocean (130–150 °W). Data was collected from up to 14 days back if the minimum amount of 10 points per 1° latitude band was not met. These regression coefficients, slope *a* and offset *b*, were used to get the bias ($B_{\text{lat}}$) for each pixel, at day t and within latitude band *lat*, based on its reflectance at 744 nm (*R*), as shown in Eq. (5).

$$B_{\text{t, lat}} = a_{\text{t,lat}} \cdot R(\lambda_{744}) + b_{\text{t,lat}} \tag{5}$$

Since the bias is additive due to the false-infilling or false-deepening, SIF is adjusted for the bias by subtracting the estimated bias, see Eq. (6).

$$I'_{\text{SIF}} = I_{\text{SIF}} - B_{\text{t, lat}} \tag{6}$$

Here we made two modifcations from the adjustment method of SIFTER v2. First, we extended the non-vegetative area to

obtain the regression coefficients from to cover a broader latitude range. Not only the Pacific Ocean (130–150 °W), but also a box over Atlantic Ocean (12°W–2°E) is used as reference area. Secondly, we relaxed the filtering criteria to include pixels with any cloud fraction to calculate the regression coefficients, instead of the <0.4 requirement as used in van Schaik et al. (2020). Accepting pixels over the reference area with any cloud fraction does not only significantly enhance the number of collected pixels, which resulted in better fits, but also enables representation of pixels with higher radiance values. Representativeness

of higher radiance values is hypothesized to help address biases over non-vegetative desert area (Joiner et al., 2016).

The light blue line in Fig. 10a represents SIF with the *post hoc* bias-adjustment, gridded per latitude band. The adjusted SIF portrays a realistic near zero-line across latitude over the Pacific ocean, indicating that the adjustment methods succeeds in removing the latitude-dependent bias. The effect of the adjustment is also noted when comparing unadjusted, Fig. 10a, and adjusted SIF, Fig. 10b. The adjusted SIF shows values around zero over the oceans, and adjusted SIF no longer suffers from

negative values over the desert areas.

Next, to evaluate the impact of the inclusion of cloudy pixels in constructing latitude bias adjustment in Eq. (5), we constructed the latitude bias correction using SIFTER v3 SIF *with* cloud fraction requirements, as in SIFTER v2. Figure S7 shows that the resulting latitude bias is more negative over the Sahara area as compared to no cloud filtering. This more negative bias correction can lead to unrealistic SIF values exceeding well over zero. It indicates that including cloudy pixels in the latitude

bias computation indeed provides more accurate adjusted SIF values, particularly over desert areas.



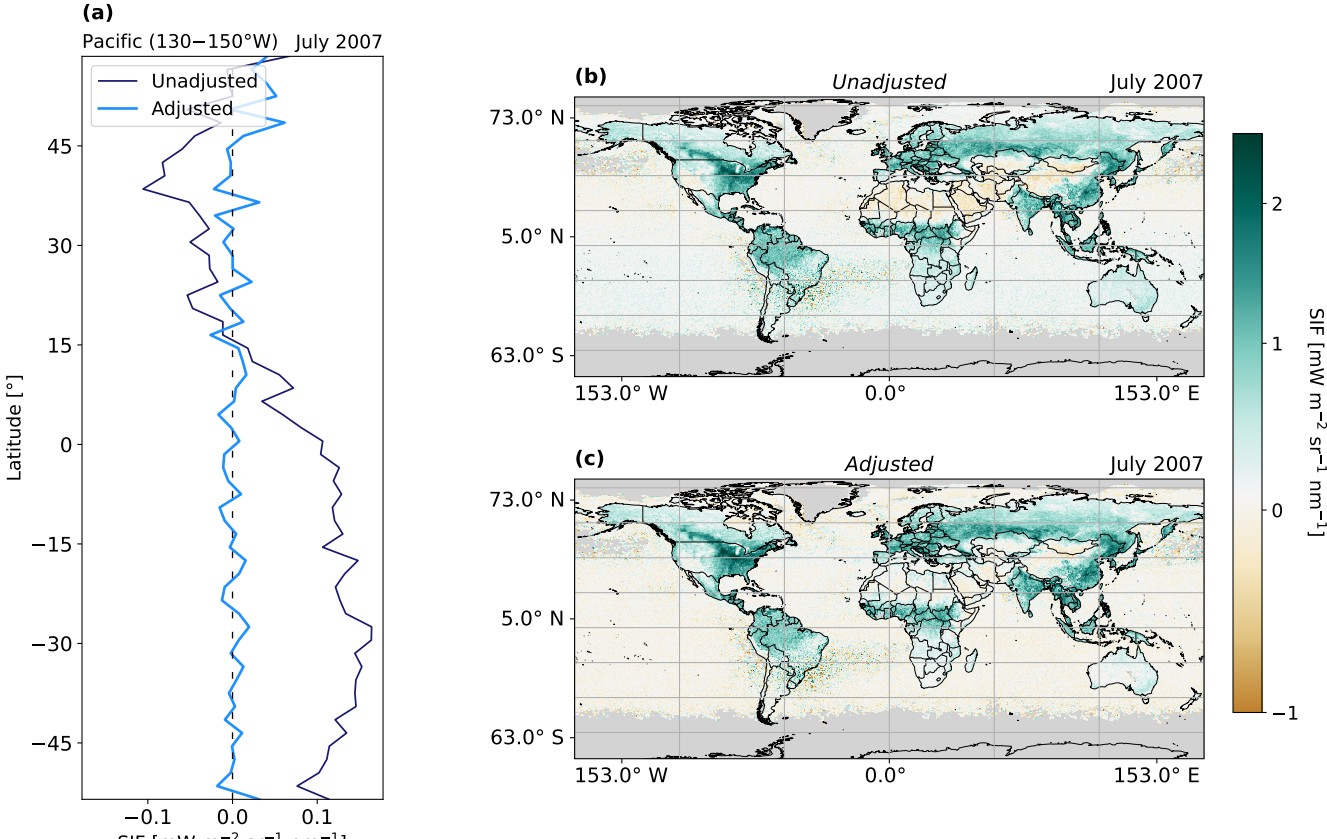

**Figure 10.** (a) Adjusted (light blue line) and unadjusted (thin dark blue line) SIF over the Pacific Ocean averaged per latitude band of 2° and over July 2007, (b) level-2 SIF averaged over July 2007 without zero-level offset correction, (c) level-2 SIF averaged over July 2007 with zero-level offset correction as post-hoc correction.

## 5 SIFTER v3 retrieval results

### 5.1 Impact of in-flight degradation correction on retrieved SIF

To examine the impact of our degradation correction on the SIF retrievals, we compared time series of SIFTER v3 SIF values retrieved from corrected and uncorrected reflectances. Both SIF values, degradation corrected and uncorrected, were adjusted for the zero-level bias. Figure 11 shows this comparison across the Sahara (unvegetated) and five diverse vegetated areas: Eastern Europe, United States Cornbelt, the Amazon, the Congo Basin, and Southeastern Australia.

In Eastern Europe, United States Cornbelt, the Amazon, and the Congo Basin, uncorrected SIF values (thin dark blue line) surpasses degradation-corrected SIF values (light blue line) before 2014. After 2014, uncorrected SIF falls below the corrected values. For instance, in 2010, the average degradation-corrected SIF over the United States Cornbelt was 0.67 mW m$^{-2}$ sr$^{-1}$





nm$^{-1}$, while the uncorrected SIF averaged 0.71 mW m$^{-2}$ sr$^{-1}$ nm$^{-1}$, representing a 2.8 % difference. By 2016, the corrected
SIF values were 5.6 % higher than the uncorrected SIF in the same region. The largest difference between the degradation-
corrected and uncorrected SIF is found over the Amazon region between 2014–2017, with a discrepancy of +11.1%. The
observed patterns indicate that degradation correction reduces SIF values in the earlier years and increases them in later years,
reflecting the inverse function of the applied degradation factors. In the absence of an appropriate degradation correction,
deviations from true reflectance values are, at least partly, being misattributed as SIF values by the retrieval algorithm.

**Figure 11.** Time series of Level 2 SIF retrieved with the SIFTER v3 algorithm with (solid light blue line) and without the in-flight degradation correction (thin dark blue line) over the Sahara (16–30°N, 8° W–29° E) and vegetated areas: Eastern Europe (50–55° N, 24–39° E), United States Cornbelt (38–46° N, 81–96° W), Amazon (0–15° S, 55–70° W), Congo Basin (13° S–6° N, 14–31° E), and Southeastern Australia (35.5–38° S, 141.5–149.5° E). Co-sampled SIF data from SIFTER v2 is plotted in a dashed orange line.

The zero-level offset adjustment, included in the time series shown in Fig. 11, could impact the consistency of SIF over time.
To isolate the effect of degradation correction from this *post hoc* correction, we compared the time series of SIFTER v3 with





and without the degradation correction for both adjusted (final product) and unadjusted values in Eastern Europe and the United
States Cornbelt, as shown in Fig. S10. We observed a clear decreasing trend in annual maximum SIF values in the uncorrected
SIF values when the zero-level offset adjustment is *not* applied; but this trend disappears when the adjustment is applied. This
suggests that the *post hoc* correction plays a significant rol in removing the degradation trend in the uncorrected SIF. This
diminishing effect of the *post hoc* correction is further emphasized by a 5.6 % and 23.3 % difference in degradation-corrected
versus uncorrected SIF over the United States Cornbelt in 2016, respectively with and without the *post hoc* zero-level offset
adjustment.

Overall, the degradation correction effectively removes spurious trends over time, resulting in more consistent SIF time
series. The full impact of our algorithm revisions on the SIF product will be assessed by comparing SIFTER v3 with the
previous dataset, SIFTER v2, in 5.2.

## 5.2    Comparison of SIFTER v3 with SIFTER v2

We compared SIF from the new SIFTER v3 with our previous product, SIFTER v2 (van Schaik et al., 2020). We co-sampled the
datasets to allow a fair comparison. The re-processed level-1b R3 dataset used in v3 includes a different version of FRESCO+,
resulting in slight changes in effective cloud fractions compared to the previous v2 (Fig. S8). Figure 11 illustrates the time series
for both SIFTER products across unvegetated Sahara area and five vegetated areas. The analysis over the Sahara, where SIF is
expected to remain near zero, serves as a validity check. The new product reflects this validity, with an average of $0.08\pm0.08$
mW m$^{-2}$ sr$^{-1}$ nm$^{-1}$, while the previous product fluctuate around $0.27\pm 0.13$ mW m$^{-2}$ sr$^{-1}$ nm$^{-1}$. This discrepancy may be
attributed to the revision in the zero-level offset adjustment calculation, as discussed in 4.3.

   Clear seasonal patterns are observed in both products across the vegetative regions. In Eastern Europe, the Amazon and
the Congo Basin, SIFTER v2 shows higher values than SIFTER v3 until around 2013, after which it drops below SIFTER
v3. For example, in 2010, v2 shows values of 1.32 mW m$^{-2}$ sr$^{-1}$ nm$^{-1}$ over the Amazon, compared to 1.21 mW m$^{-2}$ sr$^{-1}$
nm$^{-1}$ from v3 – an 8.2 % difference. By 2016, the v2 annual average is 11.3 % lower than that of the new v3 product. This
pattern aligns with trends in reflectance degradation before 2014 (see Fig. 2). A similar trend was found when comparing
degradation-corrected SIFTER v3 with the uncorrected values, as discussed in 5.1.

   Table 4 shows SIFTER v3 and SIFTER v3 values for January and July 2008 across the vegetated areas. The year 2008 was
chosen to minimize degradation-related differences. The magnitudes in SIF are similar, with a maximum difference of around
-10 % over the Amazon in January. The higher SIFTER v2 values were expected following the algorithm revisions (see Table
3). For the analyzed vegetated regions and months, v3 exhibits lower standard deviation and variability than v2. This suggests
improved precision and consistency of SIFTER v3. This is supported by the lower uncertainty in the SIFTER v3 SIF values,
which is approximately 13 % lower on January 8, 2008, and 10 % lower on July, 1, 2018, as shown in Fig. 12. These increased
precisions result from reduced fit residuals following the revisions in the SIFTER algorithm (Table 3).

   Overall, SIFTER v3 demonstrates enhanced consistency over space and time, and precision compared to SIFTER v2. How-
ever, to confirm whether the new version accurately captures the interannual variation of vegetation activity across the 2007–
2017 record, further evaluation against independent data is necessary. This will be further analyzed and discussed in 5.3.



**Table 4.** Monthly mean of co-sampled SIF values from SIFTER v3 and SIFTER v2 for the different vegetated regions as shown in Fig. 11 for January and July 2008. The uncertainties reflect the standard deviation. SIFTER data has been selected for autocorrelation values < 0.2 and cloud fraction < 0.4.

| | January 2008 | | July 2008 | |
| --- | --- | --- | --- | --- |
| | SIFTER v3 $[\mathrm{mW\ m^{-2}\ sr^{-1}\ nm^{-1}}]$ | SIFTER v2 $[\mathrm{mW\ m^{-2}\ sr^{-1}\ nm^{-1}}]$ | SIFTER v3 $[\mathrm{mW\ m^{-2}\ sr^{-1}\ nm^{-1}}]$ | SIFTER v2 $[\mathrm{mW\ m^{-2}\ sr^{-1}\ nm^{-1}}]$ |
| Eastern Europe | $0.10 \pm 0.36$ | $0.15 \pm 0.38$ | $1.20 \pm 0.61$ | $1.20 \pm 0.72$ |
| US Cornbelt | $0.12 \pm 0.44$ | $0.15 \pm 0.47$ | $1.61 \pm 0.80$ | $1.53 \pm 0.86$ |
| Amazon | $1.37 \pm 0.84$ | $1.52 \pm 0.96$ | $0.84 \pm 0.78$ | $0.90 \pm 0.86$ |
| Congo Basin | $0.84 \pm 0.65$ | $0.82 \pm 0.77$ | $0.45 \pm 0.60$ | $0.51 \pm 0.69$ |
| SE Australia | $0.48 \pm 0.57$ | $0.45 \pm 0.64$ | $0.27 \pm 0.40$ | $0.22 \pm 0.43$ |

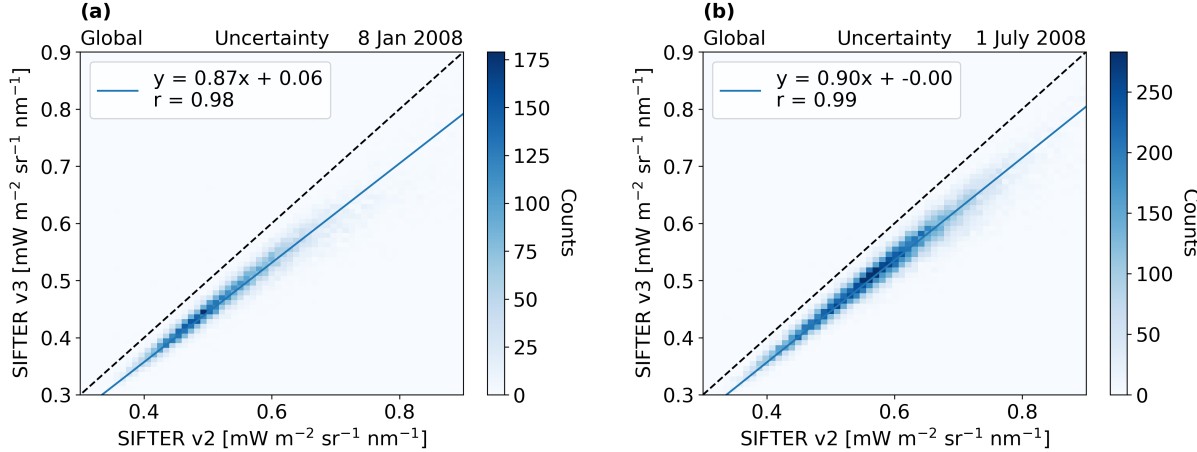

**Figure 12.** Pixel-by-pixel comparison of the uncertainty of SIFTER v3 and SIFTER v2 derived over all land-pixels on (a) 8 January 2008 and on (b) 1 July 2008. The pixels cover the global land area and are filtered for autocorrelation values <0.2 and cloud fractions <0.4.

## 5.3 Evaluation of SIFTER v3 with GPP observations

To evaluate the ability of SIFTER v3 to accurately track vegetation activity over time, we compared it with two independent datasets, namely FluxSat GPP (Joiner et al., 2018) and FLUXCOM-X (Nelson et al., 2024). Both state-of-art GPP products are widely used to track interannual variations in GPP, are known for capturing drought events (Byrne et al., 2021; Chen et al., 2021; Lv et al., 2023), and align well with GPP estimates from independent flux towers (Joiner et al., 2018; Zhang et al., 2023).

FluxSat GPP is a high-resolution, $0.05° \times 0.05°$, satellite-derived dataset obtained using a neural network approach and daily-scaled MODIS reflectance (MCD43) data to upscale GPP estimates from eddy-covariance flux towers (FLUXNET 2015). FLUXCOM-X similarly upscales GPP estimates from eddy-covariance data and daily-scaled MODIS reflectance data, but also



incorporates meteorological data from the ERA5 reanalysis products, land surface temperature observations and land cover information (Tramontana et al., 2016; Jung et al., 2019; Nelson et al., 2024). The variations in FLUXCOM-X are driven by MODIS remote sensing data (Nelson et al., 2024). This product is obtained through a machine learning approach based on gradient boosted regression trees implemented with the XGBoost library, and it has a hourly temporal scale with a spatial resolution of $0.05° \times 0.05°$ (Nelson et al., 2024).

Figure 13 shows standardized monthly averaged time series FluxSat GPP, FLUXCOM-X GPP, SIFTER v3 and SIFTER v2 for the vegetative regions shown in Fig. 11 and Table 4. The standardization adjusts the dataset in a way that the values are expressed in units of standard deviation from the mean (also see the caption of Fig. 13). This adjustment allows for direct comparison of the SIF and GPP products at the same scale. The seasonal patterns of FluxSat GPP compared well with both SIFTER products, indicating that SIFTER effectively captures the seasonal variability across different geographical

regions. FLUXCOM-X GPP also closely follows both SIFTER products and FluxSat GPP, although deviations are observed in the Amazon and Congo Basin regions, which may result from known challenges in capturing carbon fluxes over the tropics (Tramontana et al., 2016).

SIFTER v3 follows the seasonal peaks and troughs depicted by FluxSat GPP more accurately, compared to SIFTER v2. This is particularly evident in the Amazon region where SIFTER v2 shows a decreasing trend between 2014 and 2017, while

SIFTER v3 and FluxSat GPP remains more consistent over time. We found a consistent pattern of overestimation (before 2014) and underestimation (after 2014) by SIFTER v2 relative to FluxSat GPP. This pattern, also observed in Fig. 11 and discussed in Sect. 5.1 and 5.2, emphasizes that the lack of degradation correction in SIFTER v2 from 2007 to 2013 lead to an overestimation of SIF, and that the applied correction in the later years did not remove degradation effects completely.

Overall, SIFTER v3 highly correlates with FluxSat GPP (r=0.91–0.99, Fig. S13) and FLUXCOM-X GPP (r=0.85–0.99, Fig.

S14) across all regions, indicating strong consistency, both temporally as well as across space, in capturing vegetation activity over the record. Compared to SIFTER v2, SIFTER v3 shows higher correlation with both GPP products, reflecting a more robust and consistent pattern interannual variations in SIF over the 2007–2017 period. Moreover, SIFTER v3 data exhibits a significantly reduced spread in values, with measurements more tightly clustered together compared to SIFTER v2. This reduced variability indicates that the improvements in the SIFTER v3 algorithm have successfully minimized outliers, leading

to enhanced consistency and precision in the SIF dataset.

## 6 Conclusions

We improved the Solar-Induced Fluorescence of Terrestrial Ecosystems Retrieval (SIFTER) retrieval algorithm to obtain a more consistent and accurate solar-induced fluorescence (SIF) long-term record SIF record from the GOME-2A instrument for 2007–2017. Although GOME-2A data extends to 2021, measurements beyond 2018 were excluded due to orbital control

issues of Metop-A. The updated algorithm, SIFTER v3, uses the recalibrated and reprocessed GOME-2A level-1b Release 3 (R3) data as input. Additionally, it incorporates an enhanced understanding of the instrument's characteristics, leading to strong improvements in fit uncertainty, and spatial and temporal consistency.





**Figure 13.** Time series of standardized FluxSat GPP (solid black line), FLUXCOM-X GPP (dashed black line), and SIF from SIFTER v3 (solid blue line) and SIFTER v2 (dashed orange line) across the five vegetative regions of Figure 11. All data are monthly averaged. The monthly data are standardized by subtracting each value (x) by the averaged value over all months ($\mu$) and divided by the standard deviation across 2007–2017 ($\sigma$); the standardized value is calculated as x = $(x_0-\mu)/\sigma$.

We found that the level-1b R3 data is more consistent than previous level-1 version, but instrument degradation still affects the reflectance data. We found that the reflectance degradation strongly depends on time, wavelength, and scan-angle. The reflection degradation manifests already early in the record and evolves over time, in response to throughput tests, and different






trends in radiance and irradiance degradation. To address these degradation-related trends, our SIFTER v3 algorithm applies a correction that depends on time (daily time step), wavelength, and scan-angle, on the observed R3 reflectances. Compared to SIFTER v2, which only corrected for degradation from June 2014 and lacked scan-angle dependency, SIFTER v3 provides more consistent SIF records.

Other improvements made to the SIFTER algorithm include more orthogonal principal components (PCs), and a better representation of slit function variation across spectra. The latter resulted in more realistic solar irradiance $\overline{E_0(\lambda)}$ with deeper Fraunhofer lines. These algorithm revisions led to lower magnitudes in SIF and reduced fit residuals across the retrieval window. Particularly, the new PCs improved the performance over the 734–745 nm region, where water vapor features are present, lead to a ~14 % reduction in fit residuals. In SIFTER v3, the uncertainty in retrieved SIF decreased by approximately 10

% compared to SIFTER v2, which is the direct result of the reduced fit residuals. The improved temporal consistency of SIFTER v3 is most evident in tropical regions with higher atmospheric water vapor content, like the Amazon and the Congo Basin, highlighting its enhanced ability to capture the atmospheric effects on the spectra, particularly the effect of water vapor absorption.

Despite these improvements, a persistent latitude bias remained, observed as a zero-level offset over non-vegetative areas.

This zero-level offset bias is most likely due to varying slit functions in response to instrumental temperature warming across the orbit. To correct for this, we derived daily latitude-dependent adjustments over the Pacific and Atlantic Oceans. These adjustments are applied as a *post hoc* correction (additive). SIFTER v3 obtains the adjustments from all pixels – both cloudy and clear-sky – while SIFTER v2 solely used clear-sky pixels. Including cloudy pixels to obtain the zero-level adjustment enhanced the adjustment's representativeness in desert deserts. As a result, SIFTER v3 shows more realistic values of 0.08

mW m$^{-2}$ sr$^{-1}$ nm$^{-1}$ over the Sahara area, compared SIFTER v2 with less realistic SIF values of 0.27 mW m$^{-2}$ sr$^{-1}$ nm$^{-1}$.

We evaluated the new data record against independent FluxSat GPP data and FLUXCOM-X GPP data. SIFTER v3 is strongly correlated with FluxSat GPP across different biomes, and outperforms SIFTER v2. This improved performance is also observed when comparing FLUXCOM-X GPP with the new and previous SIFTER product. Throughout the entire 2007– 2017 period, SIFTER v3 showed strong temporal consistency with both GPP products, without signs of degradation induced

false-trends. Additionally, SIFTER v3 exhibited substantially reduced variability across all analyzed regions.

In conclusion, our new SIFTER v3 product resulted in more robust and accurate SIF values, with realistic ~0 values around the desert areas, and with a lower uncertainty of 10 %. Moreover, the new improved product showed temporal consistency for the entire 2007–2017 period across various biomes and regions. Compared to SIFTER v2, the values of SIFTER v3 follow the independent FluxSat and FLUXCOM-X GPP data more consistently over space and time, indicating that the SIFTER v3

product is a reliable proxy to track vegetation activity over the 2007–2017 period.

Future effort will address the use of the new SIFTER v3 algorithm to retrieve SIF records from the GOME-2B and GOME-2C instruments, that also both suffer from similar reflectance degradation as GOME-2A.



*Data availability.* The GOME-2A SIF data processed with the SIFTER v3 algorithm will become publicly available. The GOME-2 SIF data are provided by KNMI within the framework of the EUMETSAT Satellite Application Facility on Atmospheric Composition Monitoring (AC SAF).


*Author contributions.* JCSA and KFB designed the study. JCSA and ONET processed the algorithm revisions and led the SIFTER v3 reprocessing. ONET improved the slit function interpolation methodology, JCSA refined the principal component analysis, and LGT provided methodology and code to analyze the characteristics in reflectance degradation, which formed the base of our in-flight degradation correction. JCSA performed the data analysis, with KFB, LGT, and ONET in assisting with the interpretation. WWV provided the FLUXCOM-X GPP time series over the five vegetated areas. JCSA led the writing and KFB contributed to the conceptualization of the manuscript and figures, including development of the structure, content and flow, and provided revisions for intellectual content. All authors reviewed the manuscript.


*Competing interests.* One of the co-authors is a member of the editorial board of AMT.

*Acknowledgements.* The work presented in this paper was supported by the European Organization for the Exploitation of Meteorological Satellites (EUMETSAT) via the Atmospheric Composition Satellite Application Facility (AC-SAF). EUMETSAT is also acknowledged for providing the GOME-2 level-1b data (both the R3 data and the NRT data that followed). We would like to thank the European Union's Horizon 2020 research and innovations programme under grant agreement no. 869367 (EU LANDMARC project) for providing the seed funding that facilitated the initial development of this research. We further acknowledge the use of FluxSat GPP and FLUXCOM-X GPP data.




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
