# Peer review of "Improved consistency in solar-induced fluorescence retrievals from GOME-2A with the SIFTER v3 algorithm"

_EGUsphere, 2024_

## Referee Comment (RC1)

**Referee Comment for**

**egusphere-2024-2666 | Journal relation: AMT**    Submitted on 26 Aug 2024
**Improved consistency in solar-induced fluorescence retrievals from GOME-2A with the SIFTER v3 algorithm**
J. C.S. Anema, K. F. Boersma, L. G. Tilstra, O. N.E. Tuinder, and W. W. Verstraeten

Anema et al. present results from an update to their Solar-Induced Fluorescence of Terrestrial Ecosystems Retrieval (SIFTER) v3 algorithm applied to observations from the GOME-2A sensor, together with extensive comparisons to the previous SIFTER v2. Updates include the use of the latest level 1 GOME-2A radiance product, improvements to better account for instrument degradation, and changes to background ("zero offset") and latitudinal bias corrections.

Space-based measurements of Solar-Induced Fluorescence (SIF) have become established data products and are routinely observed from sensors including GOME, SCIAMACHY, GOME-2, GOSAT, OCO-2, TROPOMI, and OCO-3, with a combined data record that goes back to 1995. SIF is a highly challenging measurement to make from space, and updates and improvements to existing data products are highly welcome to reduce data uncertainty and enhance data consistency and accuracy.

Anema et al. demonstrate extensively and convincingly that their v3 SIFTER results present an improvement over the SIFTER v2 product in terms of consistency. However, they do not present any evidence about either version's accuracy. The only comparisons to non-SIFTER data are shown in Figure 13: scaled results from SIFTER are plotted against GPP measurements from FluxSat and FLUXCOM-X products to show that seasonal and inter-annual variations in GPP are reproduced by SIFTER SIF v3 better than v2. This is not evidence for the accuracy of the new version, only for its consistency.

A wide range of independent satellite-based SIF data products have been publicly released, including that of OCO-2 starting in September 2014 providing about three years of temporal overlap with GOME-2A. While the basic scope of this paper as a "algorithm modifications and product improvement" doesn't have to change, I feel strongly about the need to include, at the least, a comparison plot with independent SIF observations for a perspective on where the SIFTER results fall in relation to data from other instruments. As has been the case for a long time now with minor trace gases like BrO, H2CO, or C2H2O2, SIF is no longer a "first observation" type of measurement, and new data products should be benchmarked against published data records that have been accepted as the current standard. This is not to suggest that existing records are necessarily correct or that deviating new results are necessarily wrong. SIF in particular is a challenging observation to make, and "dissenting opinions" only help to move the state of these measurements forward. In this particular case, the seasonal peak SIF values shown in Figure 11 appear to be 20-40% higher than those reported from other satellite sensors, for essentially all vegetated regions. That warrants an explanation as to possible sources for these differences and the confidence in the results.

To enhance the scientific significance of this study, while keeping its focus as an algorithm paper, I recommend the following modifications to the manuscript:

- Streamline the discussion of the differences between v2 and v3, which can be presented in considerably abbreviated form without sacrificing insights into the modifications.

- Add a comparison plot to an independent space-based, non-SIFTER SIF data product (e.g., the biomes in Figure 11 could be augmented with data from another satellite instrument or instruments) and a brief discussion of how the data products relate to each other.

After those modifications, the paper should be submitted for re-review.

The following comments are more detailed and editorial in nature and may help the authors during the revision of the manuscript. They are mostly intended as suggestions rather than mandatory points to be addressed, though several issues will benefit from clarifications.

**Introduction**:
suggest to include this paper for OCO-2/3 SIF reference
*Global GOSAT, OCO-2, and OCO-3 solar-induced chlorophyll fluorescence datasets; R. Doughty et al., Earth Syst. Sci. Data, 14, 1513–1529, 2022* https://doi.org/10.5194/essd-14-1513-2022

**Figure 1:**
"different hues of grey" isn't working well; would suggest "orange", "light red", "dark red" or something similar, to show pre-6.3.3 processor version. Alternatively, time frames of each processor version should be included either in the figure description or indicated in the plot (shading, lines, etc.) to give the reader an idea which processor version was used when.

The mid-2013 drop must be the change in throughput related to the switch to narrow swath. But why exactly does the reflectance drop? Should that not be taken care of by updates to the radiometric calibration? The atmosphere doesn't change with the switch to a reduced swath, and vicarious calibration or cross-sensor radiometric comparisons (MODIS, etc.) should provide information on the actual radiance levels.

How do the 740 nm R3 reflectances in this figure relate to the equivalent 747 nm R3 reflectances in Figure S2 that show a smooth transition across the switch to reduced swath?

**Figure 2:**
Why are post-2013 reflectances not shown? Would it be instructive to limit pre- and post swath reduction reflectances to the extend of the reduced swath?

**Figures 3&4:**
These could be combined, since they are principally showing the same thing; as for visual cosmetics, discrete color levels (12?) might introduce some structure into the monotone Figure 4.

**Figures 6,8,9:**
Those panels could be combined into a single figure (with shared x-axes to save vertical space).

**Figure 7:**
The SIFTER 3 ILS is lost in the line width, to the point that the visual effect is somewhat strange; suggest to reduce line thickness (or switch v2 and v3 thickness), and/or include a zoom of, e.g., the 741-742 nm region.

For general information: The TSIS solar reference spectrum is becoming more widely adopted as the standard irradiance reference; absolute radiometric levels differ slightly from Chance/Kurucz (see image below). https://lasp.colorado.edu/lisird/data/tsis1_hsrs_p1nm

[Figure]

**Zero-Level Offset Adjustments:**
First, for reference: practically every existing SIF retrieval approach neglects the effect of inelastic Raman scattering on the Fraunhofer lines. This introduces an error in SIF retrievals that, while negligible over high-SIF biomes, disproportionally affects low-SIF regimes and, with that, necessarily zero-level offset corrections. A study to quantify this effect is currently under review (and thus not available to the authors of this manuscript).

SIFTER v3 switched to including fully cloudy pixels for background correction, which means more implicit variability of rotational Raman scattering in the background references. More clouds will mean less atmospheric Raman scattering, hence less reduction in Fraunhofer line depth and thus less "erroneous SIF" over non-fluorescing surfaces. Does this conform with the change in background correction values shown in Figure S7? That figure is a little hard to interpret (and it may also benefit from a tightening of the plot range to ±0.5 or ±0.4).

**Figure 10:**
Is the the latitude-dependent ILS is known, or can it be derived from in-flight spectra? Would that help with the latitude-dependent offset correction?

**General:**
Ever so often, use "allow" instead of "enable"

**Line 73:**
Check the font – is it "oh cee ell oh" or "oh cee eye oh"? (OClO or OCIO)

**Line 88:**
proceeded → performed

**Line 90:**
Are there any details on this "drop of throughput"? Specifically why does it affect the reflectances?

**Line 133:**
"narrow swath" and "nadir static" supposedly are a special observation modes?

**Equation 1:**
by itself, this doesn't provide much information that couldn't be conveyed by text alone. Can the full equation be provided?

**Line 162:**
"Scanner Angle" (and "Scanning Angle" or "Scan-Angle") → "Scan Angle"

**Line 165:**
"sensor-switch" supposedly is the change to the reduced swath? That term is a bit confusing, "swath reduction" would be better.

**Line 171:**
post-sensor switch → post sensor-switch

**Line 182:**
"relative filling-in of solar Fraunhofer absorption lines" → "reduction of solar Fraunhofer line depth in the radiance spectra"
that makes it clear in which spectra this is happening, and it also makes it intuitive that the overall effect is SFLs showing up as enhancements (peak)s in the I/I0 reflectances, from a combination of SIF and inelastic Raman scattering.

**Figure 8a:**
Absolute uncertainties remain the same, thus relative uncertainties increase by 15% - does that hold true in general?

**Line 282:**
Can a reduction from 0.068% to 0.063% in RMSE really be considered "substantial" or "significant"? While that is indeed a reduction of ~10% in relative RMSE, what are the corresponding values in terms of absolute radiance ( ~?x10-4)?

---

## Author Comment (AC2)

**Referee #2**

We thank Referee 2 for this review of our manuscript. Below, we address the comments with the comments of Referee 2 in bold and our reply in normal font.
* * *
**Overall impression**
**The manuscript proposes and describes a new processing of GOME2 data that improves the SIF retrieval. The results do show convincing improvements and the description is clear and detailed. The resulting dataset will be useful for the community and this manuscript will serve as a good reference for those who need to go in the details.**
**I am not an expert in the actual SIF retrieval nor the GOME instruments. I must admit that this manuscript is more technical than I initially thought, and that it thus fall beyond my comfort zone in terms of technical details. Therefore I cannot pronounce myself too much on the very technical satellite retrieval details and hope that this is covered by other reviewers.**
**Specific points**
Reply: We thank Referee 2 for these comments.

**L61: Maybe state that this is FLUXCOM X-BASE products**
Reply: We thank Referee 1 for this comment. It is indeed necessary to specify that we used FLUXCOM-X. We will modify line 61.

**L88: Not too sure (for me) how the information on the throughput tests is actually useful for the average reader. Maybe some more context (if needed) could help.**
Reply: To correct for instrument degradation, it is important to understand the various drivers of the varying degradation rate. The throughput tests played a significant role in the changing degradation rate over time and its scan-angle dependency. We acknowledge that this aspect is not made clear enough in line 88 and that this specific paragraph should be modified to highlight the different effects that contributed to the variability of the degradation impact.

**L116: What seems also very clear is a downward trend after the jump. This would be good to point out (and state the reasons behind)**
Reply: The clear downward trend after the jump, after ~2014, is caused by instrument degradation. This trend is mentioned in lines 115—116. We agree that it should be more explicitly stated that this downward trend is due to the instrument degradation effect. We will modify this sentence to enhance clarity.

**L140: is this assumption correct given noted trend in global greening?**
Reply: The impact of land-use change and other factors on the variation of the global reflectivity is studied intensively but its global impact is uncertain and contrasting results are found (Li et al., 2022). In relation to the impact of throughput loss due to instrument degradation, up to +10% in terms of reflectance (EUMETSAT, 2022), the reported geophysical trends due to e.g. greening are much smaller, with MODIS data estimating a global decrease in global albedo of 0.0004 between 2002 and 2016 (Li et al., 2018).

Therefore, such geophysical trend on the global reflectance over the studied period would be smaller than the accuracy of our method (Tilstra et al., 2012).

**Fig 2: To be clear, the +0.1 should also be mentioned in the legend**
Reply: We thank Referee 1 for this comment. We will include the +0.1 in the legend.

**L192: for completion, please state what E0 is in Eq. 3**
Reply: We appreciate the Referee's comment regarding the not-explained $E_0$ variable in Equation 3. $E_0$ is the solar irradiance, we will state the meaning in $E_0$ after Eq. 3.

**Fig 11: I feel this visualization does not show well the actual improvements. Consider additional/complementary plots showing residuals with respect to the mean seasonal cycle, or differences with respect to one product.**
Reply: SIF is sensitive to changing vegetation dynamics due to disturbances such as droughts, resulting in interannual variation in seasonal SIF (e.g., Koren et al., 2018). Residuals with respect to the mean seasonal cycle would therefore not be useful to diagnose improvements.

We agree with the Referee that showing the difference in SIF (SIFTER v2, and SIFTER v3 without degradation correction) over time with respect to the SIFTER v3 product will enhance the presentation of the improvements. We will include such plot in the supplement.

**Fig 12: why these two dates, which are showing very similar information? Would it not be more appropriate to show a date after the sensor jump of 2013 to see if things hold there too?**
Reply: Figure 12 shows the correlation between SIF uncertainty of SIFTER v2 and SIFTER v3 over two days in different seasons. We agree with the referee that it would be insightful to show this correlation for a later date when degradation impact is more significant.

**Fig 13: it is a pity that the spatial variability is not well showcased. Could you consider adding another figure showing differences in spatial patterns over these regions (i.e. showing the actual spatial variability with maps rather than time series)?**
Reply: We appreciate the Referee's comment regarding the lack of presenting spatial differences between the SIF products in this work. We agree that showing the spatial variability will provide more insight into the showcased regions, therefore spatial variability maps will be added to the supplement of this manuscript.

**L398: My understanding is that FluxSAT does use some GOME2 data at some point in their processing, while FLUXCOM XBase does not at all. Please investigate/confirm is this is the case and discuss the possible repercussions (and circularity) that may come out from this comparison with SIFTER.**
Reply: The Referee is right about the use of GOME-2A SIF data, specifically the SIF v27 data from NASA (Joiner et al., 2013, 2016), in FluxSat (Joiner et al., 2018). However, the NASA SIF data is only used in the FluxSat calibration procedure and not in the regression model itself, and thus we don't expect possible repercussions.

**L453: It says here the SIFTER V3 data "will become publicly available", but this does not say when. It should be made available along with this manuscript.**
Reply: The SIFTER v3 data will indeed be made available along with this manuscript.

**What about the code? It would be good practice to provide the code used to do this processing and the analyses done within this study.**
Reply: We appreciate the Referee's 2 comment. The code needed for the analysis of the SIFTER v3 L2 data will be available on request.

**References**

EUMETSAT (2022). GOME-2 Metop-A and -B FDR Product Validation Report Reprocessing R3, EUM/OPS/DOC/21/1237264.

Joiner, J., Guanter, L., Lindstrot, R., Voigt, M., Vasilkov, A. P., Middleton, E. M., ... & Frankenberg, C. (2013). Global monitoring of terrestrial chlorophyll fluorescence from moderate spectral resolution near-infrared satellite measurements: Methodology, simulations, and application to GOME-2. Atmospheric Measurement Techniques Discussions, 6(2), 3883-3930.

Joiner, J., Yoshida, Y., Guanter, L., & Middleton, E. M. (2016). New methods for the retrieval of chlorophyll red fluorescence from hyperspectral satellite instruments: simulations and application to GOME-2 and SCIAMACHY. Atmospheric Measurement Techniques, 9(8), 3939-3967.

Joiner, J., Yoshida, Y., Zhang, Y., Duveiller, G., Jung, M., Lyapustin, A., ... & Tucker, C. J. (2018). Estimation of terrestrial global gross primary production (GPP) with satellite data-driven models and eddy covariance flux data. Remote Sensing, 10(9), 1346.

Koren, G., Van Schaik, E., Araújo, A. C., Boersma, K. F., Gärtner, A., Killaars, L., ... & Peters, W. (2018). Widespread reduction in sun-induced fluorescence from the Amazon during the 2015/2016 El Niño. Philosophical Transactions of the Royal Society B: Biological Sciences, 373(1760), 20170408.

Li, Q., Ma, M., Wu, X., & Yang, H. (2018). Snow cover and vegetation-induced decrease in global albedo from 2002 to 2016. Journal of Geophysical Research: Atmospheres, 123(1), 124-138.

Li, X., Qu, Y., & Xiao, Z. (2022). Reponses of Land Surface Albedo to Global Vegetation Greening: An Analysis Using GLASS Data. Atmosphere, 14(1), 31.

Tilstra, L. G., De Graaf, M., Aben, I., & Stammes, P. (2012). In-flight degradation correction of SCIAMACHY UV reflectances and Absorbing Aerosol Index. Journal of Geophysical Research: Atmospheres, 117(D6).

---

## Author Response (AR1)

**Referee #1**

We thank the Referee 1 for this review of our manuscript. Below, we address the comments with the comments of Referee 1 in bold and our reply in normal font. Modifications to the manuscript are discussed in italic font. We numbered the comments of Referee 1.

**Anema et al. present results from an update to their Solar-Induced Fluorescence of Terrestrial Ecosystems Retrieval (SIFTER) v3 algorithm applied to observations from the GOME-2A sensor, together with extensive comparisons to the previous SIFTER v2. Updates include the use of the latest level 1 GOME-2A radiance product, improvements to better account for instrument degradation, and changes to background ("zero offset") and latitudinal bias corrections.**

**Space-based measurements of Solar-Induced Fluorescence (SIF) have become established data products and are routinely observed from sensors including GOME, SCIAMACHY, GOME-2, GOSAT, OCO-2, TROPOMI, and OCO-3, with a combined data record that goes back to 1995. SIF is a highly challenging measurement to make from space, and updates and improvements to existing data products are highly welcome to reduce data uncertainty and enhance data consistency and accuracy.**

**1. Anema et al. demonstrate extensively and convincingly that their v3 SIFTER results present an improvement over the SIFTER v2 product in terms of consistency. However, they do not present any evidence about either version's accuracy. The only comparisons to non-SIFTER data are shown in Figure 13: scaled results from SIFTER are plotted against GPP measurements from FluxSat and FLUXCOM-X products to show that seasonal and inter-annual variations in GPP are reproduced by SIFTER SIF v3 better than v2. This is not evidence for the accuracy of the new version, only for its consistency.**

Reply: The quantification of the accuracy of satellite-based SIF product is a well-known limitation and challenge. The difficulty of determining the accuracy of satellite-based SIF products is due to the lack of sufficient independent SIF observations from in-situ and airborne sensors (Rossini et al., 2022). Even with sufficient independent SIF observation, the scale mismatch between satellite observations and in-situ measurements further complicates accuracy assessment. Therefore, the validation of satellite-based SIF products primarily relies on the comparison with other SIF products. However, such cross-product comparisons are also complex as SIF is not measured directly and are further affected by uncertainties emerging from instrumental and retrieval algorithm differences.

**2. A wide range of independent satellite-based SIF data products have been publicly released, including that of OCO-2 starting in September 2014 providing about three years of temporal overlap with GOME-2A. While the basic scope of this paper as a "algorithm modifications and product improvement" doesn't have to change, I feel strongly about the need to include, at the least, a comparison plot with independent SIF observations for a perspective on where the SIFTER results fall in relation to data from other instruments. As has been the case for a long time now with minor trace gases like BrO, H2CO, or C2H2O2, SIF is no longer a "first observation" type of measurement, and new data products should be benchmarked against published data records that have been accepted as the current standard.**

Reply: We agree with the referee that new data products should be benchmarked against published data records. Our previous dataset, SIFTER v2, was compared to independent SIF data from NASA v28 in van Schaik et al. (2020), showing good consistency, both temporally and spatially. Furthermore, based on tests and fruitful scientific investigations done with SIFTER v2 in Mengistu et al. (2021), Wang et al. (2020), Wang et al. (2022), and Fancourt et al. (2022), we feel that substantial benchmarking has already taken place, and that SIFTER v2, and therefore SIFTER v3, can be considered part of the "current standard". We discuss the comparison of SIFTER v3 against other SIF data further in our reply to comment 5.

**3. This is not to suggest that existing records are necessarily correct or that deviating new results are necessarily wrong. SIF in particular is a challenging observation to make, and "dissenting opinions" only help to move the**

**state of these measurements forward. In this particular case, the seasonal peak SIF values shown in Figure 11 appear to be 20-40% higher than those reported from other satellite sensors, for essentially all vegetated regions. That warrants an explanation as to possible sources for these differences and the confidence in the results.**

Reply: We thank the referee for this comment and agree that this manuscript will benefit from a comparison against different SIF products, including an explanation regarding differences in SIF. We discuss this further in our reply to comment 5.

It should be noted that the SIF values presented in this manuscript reflect the instantaneous SIF values, rather than the daily-scaled values that are commonly presented in other work, such as Wen et al. (2020), which analyzed similar geographical regions. For GOME-2A, with a local overpass at 10:30A M, daily-scaled values are approximately 30% lower than the instantaneous values.

Modifications: *To reduce confusion regarding the SIF values shown in our manuscript, we now emphasize that the SIF values reflect instantaneous values, specifically in the captions of Figures 8, 9, S11—S20, and Table 4 of the revised manuscript.*

**To enhance the scientific significance of this study, while keeping its focus as an algorithm paper, I recommend the following modifications to the manuscript:**

**4. Streamline the discussion of the differences between v2 and v3, which can be presented in considerably abbreviated form without sacrificing insights into the modifications.**

Reply: We agree with the Referee that the discussion on the differences between v2 and v3 can be more concise.

Modifications: *We revised the "4.2 Processing and improvements of SIFTER v3 retrieval" section. This section is made mode clear by introducing all the changes, done in line 232—275 of the revised manuscript with tracked changes, before discussing and showing the impact of each individual change on the retrieval, done in lines 277—301 of the revised manuscript with tracked changes. Additionally, we combined Figures 6, 8, and 9 of the original manuscript, as suggested by Referee 1 in comment 12, see Figure 7 of the revised manuscript.*

**5. Add a comparison plot to an independent space-based, non-SIFTER SIF data product (e.g., the biomes in Figure 11 could be augmented with data from another satellite instrument or instruments) and a brief discussion of how the data products relate to each other.**

Reply: We thank the referee for this comment and agree that the comparison of the SIFTER v3 product against other non-SIFTER SIF data will provide valuable insights into the performance of the new dataset. Differences in absolute SIF values across products and instrument arise from instrumental characteristics, such as overpass time, viewing geometry, spectral and spatial resolution, and sampling, as well as differences in retrieval settings, such as the retrieval window. This has been discussed in detail in Parazoo et al. (2020), who compares different SIF products from a variety of sensors (e.g. TROPOMI, GOME-2, SCIAMACHY, GOME, and OCO-2), and discussed their differences.

The below list analyzes which SIF products are potentially suitable and which are less appropriate (or out of scope) to compare against our SIFTER v3.

Appropriate:
- GOME-2A SIF, specifically the GOME-2A NASA product (Joiner et al., 2013, 2016): this comparison is relevant as it limits differences in absolute SIF due to retrieval window setting (both use 734—758 nm) and allows for a "fair" comparison. Furthermore, it provides insight into the impact of our efforts to constrain temporal consistency.
- SCIAMACHY: SCIAMACHY: Comparison against SCIAMACHY SIF data offers a sufficient temporal overlap with GOME-2 of 5 years (2007—2011). The morning overpass of both SCIAMACHY and GOME-2A, as well as similar footprint size, respectively 30x60 km2 and 40x80 km2 over the overlapping period, allow for valuable comparison.

Less appropriate:
- OCO-2 and GOSAT SIF: The restricted global mapping of these sensors limit comparison with GOME-2A SIF data (with much larger spatial resolution) on a monthly and regional scale.

- TROPOMI SIF: Since we only retrieved GOME-2A SIF over the pre-drift period (2007—2017), there is no overlap with TROPOMI SIF data.

Modifications: *Based on the above considerations, we selected the latest version of GOME-2A SIF from NASA (Joiner et al., 2023) as the independent SIF dataset to compare against SIFTER v3 in the revised manuscript. We included the comparison of NASA GOME-2A SIF with SIFTER v3 in section 5.3, which is now titled "Evaluation of SIFTER v3 with independent SIF and GPP observations". Furthermore, we incorporated the NASA SIF time series in Figure 11 of the revised manuscript.*

*Lines 416—419 of the revised manuscript with tracked changes discusses the NASA GOME-2A SIF dataset and the relation and differences to the SIFTER v3 product. Lines 436—443 of the revised manuscript with tracked changes discusses the correlation of SIFTER v3, the two GPP products and NASA SIF. Additional figures that show the comparison between NASA SIF and SIFTER v3, as well as comparisons between NASA SIF and FluxSat GPP and FLUXCOM-X GPP are added in the revised supplementary, specifically Figures S17—S20.*

*The results of this additional analysis are discussed in the conclusion in lines 493—494 of the revised manuscript with tracked changes, and in the abstract in line 15 of the revised manuscript with tracked changes.*

**The following comments are more detailed and editorial in nature and may help the authors during the revision of the manuscript. They are mostly intended as suggestions rather than mandatory points to be addressed, though several issues will benefit from clarifications.**

**6. Introduction: suggest to include this paper for OCO-2/3 SIF reference**
***Global GOSAT, OCO-2, and OCO-3 solar-induced chlorophyll fluorescence datasets; R. Doughty et al., Earth Syst. Sci. Data, 14, 1513–1529, 2022*** https://doi.org/10.5194/essd-14-1513-2022
Reply: We thank the Referee for this suggestion. Including this reference in the introduction.

Modifications: *We now mentioned the SIF retrieval from OCO-3 and included the suggested references in line 22 of the revised manuscript with tracked changes.*

**7. Figure 1: "different hues of grey" isn't working well; would suggest "orange", "light red", "dark red" or something similar, to show pre-6.3.3 processor version. Alternatively, time frames of each processor version should be included either in the figure description or indicated in the plot (shading, lines, etc.) to give the reader an idea which processor version was used when.**
Reply: We appreciate the Referee's comment on the clarity of Figure 1.

Modifications: *We changed the colors representing the processor versions v5.3, v6.0, and v6.1 from different hues of grey to different hues of red in the revised manuscript, Figure 1 of the revised manuscript.*

**8. The mid-2013 drop must be the change in throughput related to the switch to narrow swath. But why exactly does the reflectance drop? Should that not be taken care of by updates to the radiometric calibration? The atmosphere doesn't change with the switch to a reduced swath, and vicarious calibration or cross-sensor radiometric comparisons (MODIS, etc.) should provide information on the actual radiance levels.**
Reply: The mid-2013 drop is indeed caused by the switch to the reduced 960-km swath (see lines 114-116). The reduction in swath led to observations being made at a smaller range of viewing zenith angles (from between ±55° to ±35°). These more nadir-like observations result in decreased reflectance due the bidirectional reflectance distribution function (BRDF) of the Earth's surface, which causes reflectance to vary with viewing geometry (Tilstra et al., 2021).

Modifications: *Modifications done in lines 119—121 of the revised manuscript with tracked changes clarify the reduced reflectance after July 2013.*

**9. How do the 740 nm R3 reflectances in this figure relate to the equivalent 747 nm R3 reflectances in Figure S2 that show a smooth transition across the switch to reduced swath?**

Reply: We acknowledge that the inconsistency in respective wavelength of the reflectance shown in Figure 1 and Figure S2 is confusing. The effect of changed viewing geometry, due to the swath reduction, on the reflectance differs with wavelength.

Modifications: *Figure S3 of the revised supplementary (Fig. S2 of the original supplementary) now shows the degradation corrected reflectance at 739.9 nm, the transition is similar in nature as seen in Figure 1.*

**10. Figure 2: Why are post-2013 reflectances not shown? Would it be instructive to limit pre- and post swath reduction reflectances to the extend of the reduced swath?**
Reply: To obtain correction factors for the reflectances for the entire 2007-2017 record, we used two fit periods: 2007-2012 and 2007-2017 (see Table 1) that account for the change in viewing geometry due to the swath reduction in 2012. Figure 2 shows the fits across the 2007-2012 period.

Modifications: *The correction factors for 2007—2017 (under reduced swath settings) are now shown in the supplement (Figure S2). We refer to this figure in line 159—160 of the revised manuscript with tracked changes.*

**11. Figures 3&4: These could be combined, since they are principally showing the same thing; as for visual cosmetics, discrete color levels (12?) might introduce some structure into the monotone Figure 4.**
Reply: Although Figures 3 and 4 both show the correction factors over time, they both do show different elements. Figure 3 shows the scan-angle dependency, whereas Figure 4 shows the wavelength dependency. Therefore, they could not be combined.

**12. Figures 6,8,9: Those panels could be combined into a single figure (with shared x-axes to save vertical space).**
Reply: we thank the referee for this comment.

Modifications: *We combined figure 6, 8, and 9 of the original manuscript into a single figure: Figure 7 of the revised manuscript.*

**13. Figure 7: The SIFTER 3 ILS is lost in the line width, to the point that the visual effect is somewhat strange; suggest to reduce line thickness (or switch v2 and v3 thickness), and/or include a zoom of, e.g., the 741-742 nm region.**
Reply: We thank the referee for the suggestion of including a zoom subplot.

Modification: *We altered Figure 7 of the original document (now Figure 6 of the revised manuscript). The infilling of the difference between irradiances of SIFTER v2 and SIFTER v3 (in blue) is removed, to more clearly show the individual irradiance values. Furthermore, we included a zoom of the 749—752 nm region showing the deepened Fraunhofer lines structures as seen by GOME-2A when using the SIFTER v3 algorithm in comparison to SIFTER v2.*

**For general information: The TSIS solar reference spectrum is becoming more widely adopted as the standard irradiance reference; absolute radiometric levels differ slightly from Chance/Kurucz (see image below).**
https://lasp.colorado.edu/lisird/data/tsis1_hsrs_p1nm

[Figure]

Reply: Thanks for pointing us to this improved solar reference spectrum. We intend to study the impact of this improvement on our SIF-retrievals in future studies.

**Zero-Level Offset Adjustments:**
First, for reference: practically every existing SIF retrieval approach neglects the effect of inelastic Raman scattering on the Fraunhofer lines. This introduces an error in SIF retrievals that, while negligible over high-SIF biomes, disproportionally affects low-SIF regimes and, with that, necessarily zero-level offset corrections. A study to quantify this effect is currently under review (and thus not available to the authors of this manuscript).
Reply: Thanks for making us aware of this development.

**14. SIFTER v3 switched to including fully cloudy pixels for background correction, which means more implicit variability of rotational Raman scattering in the background references. More clouds will mean less atmospheric Raman scattering, hence less reduction in Fraunhofer line depth and thus less "erroneous SIF" over non-fluorescing surfaces. Does this conform with the change in background correction values shown in Figure S7? That figure is a little hard to interpret (and it may also benefit from a tightening of the plot range to ±0.5 or ±0.4).**
Reply: Considering cloudy pixels to obtain the zero-level adjustments indeed lead to fewer "false" positive SIF over non-vegetative surfaces (lines 321—325). Thank you for the suggestion to tighten the plot range. The current figure already demonstrates the impact of no cloud filtering in comparison to cloud filtering (cf<0.4) on the bias correction clearly, therefore we decided to not change it.

**15. Figure 10: Is the the latitude-dependent ILS is known, or can it be derived from in-flight spectra? Would that help with the latitude-dependent offset correction?**
Reply: The exact latitude-dependent varying slit function is not exactly known as this effect is thought to arrange from multiple sources (Joiner et al., 2012), with the thermal instability across orbit as one of the prominent causes. Understanding the latitude-dependence of the ILS will certainly help, but by itself is not sufficient for a comprehensive correction. This is because the measurements of GOME-2A with its ILS over our reference area (the Sahara desert, 5°-25° N) are not necessarily representative for other latitudes. We are considering testing whether additional reference areas (at other latitude zones) are helping to reduce the latitude-dependent offsets.

**General:**
**16. Ever so often, use "allow" instead of "enable"**
Reply: we appreciate this comment regarding the over-use of "enable".

Modifications: *We modified line 65 of the revised manuscript with tracked changes.*

**17. Line 73: Check the font – is it "oh cee ell oh" or "oh cee eye oh"? (OClO or OCIO)**
Reply: We thank the Referee for his attentiveness. OClO is chlorine dioxide and is written as OClO (with 'ell', not 'eye').

**18. Line 88: proceeded → performed**
Modifications: *We changed "proceeded" to "performed" in line 88 of the original manuscript (line 89 of the revised manuscript with tracked changes).*

**19. Line 90: Are there any details on this "drop of throughput"? Specifically why does it affect the reflectances?**
The throughput tests concerned temporary substantial increases of the instrument's temperature. The changed detector temperatures during the second throughput test unexpectedly resulted in a loss of signal of the solar measurements (EUMETSAT, 2009, 2022). The reflectance is affected when the throughput for the earthshine optical path and solar optical path changes in a different way (EUMETSAT, 2022). Hypothetically, when they both change in the same way, the reflectance would not be affected.

**20. Line 133: "narrow swath" and "nadir static" supposedly are a special observation modes?**
Reply: Observations under narrow swath are done under a swath of 320 km, instead of the default 1920 km swath (Munro et al., 2016). For one day in each 29-day observation cycle, GOME-2A observes under narrow swath. Nadir static is an observation mode where the scan mirror points nadir without scanning, and which is used for monthly calibration of the instrument (Munro et al., 2016). The different viewing geometries of these two special observation modes result in deviating reflectance in comparison to the observations done under nominal swath (that are used for the SIF retrieval), therefore we excluded observations under narrow swath and nadir static from our analysis.

**21. Equation 1: by itself, this doesn't provide much information that couldn't be conveyed by text alone. Can the full equation be provided?**

Reply: The full equation is given in Tilstra et al. (2012). We model the global mean reflectance ($R^*$) by:

$$R^*_{\lambda,s}(t) = P^p_{\lambda,s}(t)\left[1 + F^q_{\lambda,s}(t)\right]$$

P$^p_{\lambda,s}$ is the polynomial with degree $p$, describing the long-term trend in $R^*$, and F$^q_{\lambda,s}$ is a finite Fourier series with order $q$, describing the seasonal variation. $\lambda$ is the wavelength, $s$ the scan angle and $t$ is time.

With:

$$P^p_{\lambda,s}(t) = \sum_{m=0}^{p} u^m_{\lambda,s} t^m$$

And

$$F^p_{\lambda,s}(t) = \sum_{n=1}^{q} \left[v^n_{\lambda,s}\cos(2\pi nt) + w^n_{\lambda,s}\sin(2\pi nt)\right]$$

Modifications: *We now included the full equation in the revised manuscript, specifically in equations 1—3.*

**22. Line 162: "Scanner Angle" (and "Scanning Angle" or "Scan-Angle") → "Scan Angle"**

Reply: We thank the referee for this comment. We indeed use different variations throughout the original manuscript.

Modifications: *In the revised manuscript we now consistently use "scan angle", see modification at e.g. lines 10, 56—57, 82, 174, 178, 179, the caption and heading of Table 1, and the captions of Fig. 4 and Table 2 of the revised manuscript with tracked changes.*

**23. Line 165: "sensor-switch" supposedly is the change to the reduced swath? That term is a bit confusing, "swath reduction" would be better.**

Reply: The "sensor switch" indeed refers to the reduction of the swath.

Modifications: *We now consistently refer to the 15-July event as the "swath reduction" throughout the revised manuscript, see e.g. lines 153, 177, 184—185 and 218, and the caption of Fig. 3 of the revised manuscript with tracked changes.*

**24. Line 171: post-sensor switch→post sensor-switch**

Reply: See our comment on point 23.

**25. Line 182: "relative filling-in of solar Fraunhofer absorption lines" → "reduction of solar Fraunhofer line depth in the radiance spectra" that makes it clear in which spectra this is happening, and it also makes it intuitive that the overall effect is SFLs showing up as enhancements (peak)s in the I/I0 reflectances, from a combination of SIF and inelastic Raman scattering.**

Reply: The word "relative" is of importance in line 182 of the original manuscript. With the re-emission of chlorophyll fluorescence by vegetation the radiance spectra are enhanced both inside and outside the solar Fraunhofer lines. However, due to the low radiance value within the absorption lines, the addition due to fluorescence leads to a noticeable higher relative enhancement within the solar Fraunhofer lines than outside these lines.

Modifications: *We included "in the radiance spectra" in this specific line, line 195 of the revised manuscript with tracked changes.*

**26. Figure 8a: Absolute uncertainties remain the same, thus relative uncertainties increase by 15% - does that hold true in general?**

Reply: Due to changed slit function in SIFTER v3, the absolute values of SIF decreased with respect to SIFTER v2. Figure 12 of the original manuscript shows the correlation between SIF uncertainty in SIFTER v2 vs SIFTER v3. The correlation between absolute SIF values between SIFTER v2 and SIFTER v3 is shown in Figure S13 of the supplement. The decrease in absolute SIF values is not higher than the decrease in the SIF uncertainties, therefore we don't expect a significant change in the relative SIF uncertainties.

**27. Line 282: Can a reduction from 0.068% to 0.063% in RMSE really be considered "substantial" or "significant"? While that is indeed a reduction of ~10% in relative RMSE, what are the corresponding values in terms of absolute radiance ( ~?x10-4)?**

Reply: We agree with the referee that the word "substantial" is slightly excessive. However, considering the sensitivity of the SIF retrieval an improvement of 0.063% to 0.068% should not be overlooked. The effect of this improvement is also noted as a decrease on the order of 10% in absolute SIF uncertainty (Figure 10 of the revised manuscript).

**Referee #2**

We thank Referee 2 for this review of our manuscript. Below, we address the comments with the comments of Referee 2 in bold and our reply in normal font. Modifications to the manuscript are discussed in italic font.
* * *
**Overall impression**
The manuscript proposes and describes a new processing of GOME2 data that improves the SIF retrieval. The results do show convincing improvements and the description is clear and detailed. The resulting dataset will be useful for the community and this manuscript will serve as a good reference for those who need to go in the details.
I am not an expert in the actual SIF retrieval nor the GOME instruments. I must admit that this manuscript is more technical than I initially thought, and that it thus fall beyond my comfort zone in terms of technical details. Therefore I cannot pronounce myself too much on the very technical satellite retrieval details and hope that this is covered by other reviewers.
**Specific points**
Reply: We thank Referee 2 for these comments.

**L61: Maybe state that this is FLUXCOM X-BASE products**
Reply: We thank Referee 1 for this comment. It is indeed necessary to specify that we used FLUXCOM-X.

Modifications: *We now specified the specific use of FLUXCOM X-BASE in the introduction of the revised manuscript, see line 61 of the revised manuscript with tracked changes.*

**L88: Not too sure (for me) how the information on the throughput tests is actually useful for the average reader. Maybe some more context (if needed) could help.**
Reply: To correct for instrument degradation, it is important to understand the various drivers of the varying degradation rate. The throughput tests played a significant role in the changing degradation rate over time and its scan-angle dependency.

Modifications: *In the revised manuscript, we have made modifications to better highlight the importance of understanding the impact of these throughput tests in correcting instrument degradation, see lines 88—93 of the revised manuscript with tracked changes.*

**L116: What seems also very clear is a downward trend after the jump. This would be good to point out (and state the reasons behind)**
Reply: The clear downward trend after the jump, after ~2014, is caused by instrument degradation. This trend is mentioned in lines 115—116 of the original manuscript. We agree that it should be more explicitly stated that this downward trend is due to the instrument degradation effect.

Modifications: *Modifications are made that explain the downward trend in reflectance, specifically in lines 119—121 of the revised manuscript with tracked changes.*

**L140: is this assumption correct given noted trend in global greening?**

Reply: The impact of land-use changes and other factors on the variation of the global reflectivity is studied intensively but its global impact is uncertain and contrasting results are found (Li et al., 2022). In relation to the impact of throughput loss due to instrument degradation, up to +10% in terms of reflectance (EUMETSAT, 2022), the reported geophysical trends due to e.g. greening are much smaller, with MODIS data estimating a global decrease in global albedo of 0.0004 between 2002 and 2016 (Li et al., 2018). Therefore, such geophysical trend on the global reflectance over the studied period would be smaller than the accuracy of our method (Tilstra et al., 2012).

**Fig 2: To be clear, the +0.1 should also be mentioned in the legend**
Reply: We thank Referee 1 for this comment.

Modifications: *We included the offset of +0.1 in the legend of Figure 2 in the revised manuscript.*

**L192: for completion, please state what E0 is in Eq. 3**
Reply: We appreciate the Referee's comment regarding the not-explained $E_0$ variable in Equation 3. $E_0$ is the solar irradiance.

Modifications: *In the revised manuscript, the meaning of $E_0$ is stated after Eq. 5 (was equation 3 in the original manuscript) in line 206 of the revised manuscript with tracked changes.*

**Fig 11: I feel this visualization does not show well the actual improvements. Consider additional/complementary plots showing residuals with respect to the mean seasonal cycle, or differences with respect to one product.**
Reply: SIF is sensitive to changing vegetation dynamics due to disturbances such as droughts, resulting in interannual variation in seasonal SIF (e.g., Koren et al., 2018). Residuals with respect to the mean seasonal cycle would therefore not be useful to diagnose improvements.

Modifications: *We included time series plots that show the differences in SIF between SIFTER v3 and SIFTER v2 and SIFTER v3 without degradation correction in the supplementary as Figure S19. Furthermore, as suggested by Referee 1, we included the comparison against a non-SIFTER SIF product: NASA GOME-2A SIF (Joiner et al., 2023).*

**Fig 12: why these two dates, which are showing very similar information? Would it not be more appropriate to show a date after the sensor jump of 2013 to see if things hold there too?**
Reply: Figure 12 of the original manuscript shows the correlation between SIF uncertainty of SIFTER v2 and SIFTER v3 over two days in different seasons. We agree with the referee that it would be insightful to show this correlation for a later date when degradation impact is more significant.

Modifications: *In the revised manuscript, Figure 10, we now showcase the scatter plot of SIF uncertainty from SIFTER v2 and SIFTER v3 over 8 January 2008 and 8 January 2016. Lines 403—405 of the revised manuscript with track changes discuss the consistency of the lower SIF uncertainty from SIFTER v3 over different times.*

**Fig 13: it is a pity that the spatial variability is not well showcased. Could you consider adding another figure showing differences in spatial patterns over these regions (i.e. showing the actual spatial variability with maps rather than time series)?**
Reply: We appreciate the Referee's comment regarding the lack of presenting spatial differences between the SIF products in this work.

Modifications: *We included spatial variability maps showing co-sampled daily data of SIFTER v3 and SIFTER v2 across the five vegetative regions analyzed in the supplement (Fig. S14 of the revised supplementary).*

**L398: My understanding is that FluxSAT does use some GOME2 data at some point in their processing, while FLUXCOM XBase does not at all. Please investigate/confirm is this is the case and discuss the possible repercussions (and circularity) that may come out from this comparison with SIFTER.**
Reply: The Referee is right about the use of GOME-2A SIF data, specifically the SIF v27 data from NASA (Joiner et al., 2013, 2016), in FluxSat (Joiner et al., 2018). However, the NASA SIF data is only used in the FluxSat calibration procedure and not in the regression model itself, and thus we don't expect possible repercussions.

**L453: It says here the SIFTER V3 data "will become publicly available", but this does not say when. It should be made available along with this manuscript.**
Reply: The SIFTER v3 data will indeed be made available along with this manuscript.

Modifications: *This is now stated in the revised manuscript under "Code and data availability".*

**What about the code? It would be good practice to provide the code used to do this processing and the analyses done within this study.**
Reply: We appreciate the Referee's 2 comment. The code needed for the analysis of the SIFTER v3 L2 data will be available on request.

Modifications: *This is now stated in the revised manuscript "Code and data availability".*

**References**

EUMETSAT (2009). GOME-2 FM3 Long-Term In-Orbit Degradation - Status After 1st Throughput Test, EUM/OPS-EPS/TEN/08/0588.

EUMETSAT (2022). GOME-2 Metop-A and -B FDR Product Validation Report Reprocessing R3, EUM/OPS/DOC/21/1237264.

Fancourt, M., Ziv, G., Boersma, K. F., Tavares, J., Wang, Y., & Galbraith, D. (2022). Background climate conditions regulated the photosynthetic response of Amazon forests to the 2015/2016 El Nino-Southern Oscillation event. Communications Earth & Environment, 3(1), 209.

Joiner, J., Yoshida, Y., Vasilkov, A. P., Middleton, E. M., Campbell, P. K. E., Yoshida, Y., & Huze, A. (2012). Filling-in of near-infrared solar lines by terrestrial fluorescence and other geophysical effects: simulations and space-based observations from SCIAMACHY and GOSAT. Atmospheric Measurement Techniques, 5(GSFC-E-DAA-TN9416).

Joiner, J., Guanter, L., Lindstrot, R., Voigt, M., Vasilkov, A. P., Middleton, E. M., ... & Frankenberg, C. (2013). Global monitoring of terrestrial chlorophyll fluorescence from moderate spectral resolution near-infrared satellite measurements: Methodology, simulations, and application to GOME-2. Atmospheric Measurement Techniques Discussions, 6(2), 3883-3930.

Joiner, J., Yoshida, Y., Guanter, L., & Middleton, E. M. (2016). New methods for the retrieval of chlorophyll red fluorescence from hyperspectral satellite instruments: simulations and application to GOME-2 and SCIAMACHY. Atmospheric Measurement Techniques, 9(8), 3939-3967.

Joiner, J., Yoshida, Y., Zhang, Y., Duveiller, G., Jung, M., Lyapustin, A., ... & Tucker, C. J. (2018). Estimation of terrestrial global gross primary production (GPP) with satellite data-driven models and eddy covariance flux data. Remote Sensing, 10(9), 1346.

Joiner, J., Y. Yoshida, P. Koehler, C. Frankenberg, and N.C. Parazoo. (2023). L2 Daily Solar-Induced Fluorescence (SIF) from MetOp-A GOME-2, 2007-2018, V2. ORNL DAAC, Oak Ridge, Tennessee, USA. https://doi.org/10.3334/ORNLDAAC/2292

Koren, G., Van Schaik, E., Araújo, A. C., Boersma, K. F., Gärtner, A., Killaars, L., ... & Peters, W. (2018). Widespread reduction in sun-induced fluorescence from the Amazon during the 2015/2016 El Niño. Philosophical Transactions of the Royal Society B: Biological Sciences, 373(1760), 20170408.

Li, Q., Ma, M., Wu, X., & Yang, H. (2018). Snow cover and vegetation-induced decrease in global albedo from 2002 to 2016. Journal of Geophysical Research: Atmospheres, 123(1), 124-138.

Li, X., Qu, Y., & Xiao, Z. (2022). Reponses of Land Surface Albedo to Global Vegetation Greening: An Analysis Using GLASS Data. Atmosphere, 14(1), 31.

Mengistu, A. G., Mengistu Tsidu, G., Koren, G., Kooreman, M. L., Boersma, K. F., Tagesson, T., ... & Peters, W. (2021). Sun-induced fluorescence and near-infrared reflectance of vegetation track the seasonal dynamics of gross primary production over Africa. Biogeosciences, 18(9), 2843-2857.

Munro, R., Lang, R., Klaes, D., Poli, G., Retscher, C., Lindstrot, R., ... & Eisinger, M. (2016). The GOME-2 instrument on the Metop series of satellites: instrument design, calibration, and level 1 data processing–an overview. Atmospheric Measurement Techniques, 9(3), 1279-1301.

Parazoo, N. C., Frankenberg, C., Köhler, P., Joiner, J., Yoshida, Y., Magney, T., ... & Yadav, V. (2019). Towards a harmonized long-term spaceborne record of far-red solar-induced fluorescence. Journal of Geophysical Research: Biogeosciences, 124(8), 2518-2539.

Rossini, M., Celesti, M., Bramati, G., Migliavacca, M., Cogliati, S., Rascher, U., & Colombo, R. (2022). Evaluation of the spatial representativeness of in situ SIF observations for the validation of medium-resolution satellite SIF products. *Remote Sensing*, *14*(20), 5107.

Van Schaik, E., Kooreman, M. L., Stammes, P., Tilstra, L. G., Tuinder, O. N., Sanders, A. F., ... & Boersma, K. F. (2020). Improved SIFTER v2 algorithm for long-term GOME-2A satellite retrievals of fluorescence with a correction for instrument degradation. Atmospheric Measurement Techniques, 13(8), 4295-4315.

Tilstra, L. G., De Graaf, M., Aben, I., & Stammes, P. (2012). In-flight degradation correction of SCIAMACHY UV reflectances and Absorbing Aerosol Index. Journal of Geophysical Research: Atmospheres, 117(D6).

Tilstra, L. G., Tuinder, O. N., Wang, P., & Stammes, P. (2021). Directionally dependent Lambertian-equivalent reflectivity (DLER) of the Earth's surface measured by the GOME-2 satellite instruments. Atmospheric Measurement Techniques, 14(6), 4219-4238.

Wang, S., Zhang, Y., Ju, W., Porcar-Castell, A., Ye, S., Zhang, Z., ... & Boersma, K. F. (2020). Warmer spring alleviated the impacts of 2018 European summer heatwave and drought on vegetation photosynthesis. *Agricultural and Forest Meteorology*, *295*, 108195.

Wang, S., Zhang, Y., Ju, W., Wu, M., Liu, L., He, W., & Penuelas, J. (2022). Temporally corrected long-term satellite solar-induced fluorescence leads to improved estimation of global trends in vegetation photosynthesis during 1995–2018. ISPRS Journal of Photogrammetry and Remote Sensing, 194, 222-234.

Wen, J., Köhler, P., Duveiller, G., Parazoo, N. C., Magney, T. S., Hooker, G., ... & Sun, Y. (2020). A framework for harmonizing multiple satellite instruments to generate a long-term global high spatial-resolution solar-induced chlorophyll fluorescence (SIF). Remote Sensing of Environment, 239, 111644.

Dear editor,

When examining our plotting code to combine Figures 6, 8, and 9 of the original manuscript to one single figure (Figure 7 of the revised manuscript), we identified a mistake in the printed $RMSE_{734—758}$ of Figure 8(b) and Figure 9(b). The printed $RMSE_{734—758}$ (as text) within the graph shows the values of a different pixel than of which the data is shown. The revised Figure 7 now illustrates the correct RMSE corresponding to the plotted data. The shown data is not changed, the change only concerns the printed text within the graphs.